# CDK5RAP3 Deficiency Is Associated with Hepatic Inflammation and Increased Expression of NLRP3 Inflammasome Components

**DOI:** 10.3390/biomedicines13082030

**Published:** 2025-08-21

**Authors:** Xinjin Chen, Yaqi Huang, Yilin Wu, Le Sheng, Hongchen Yan, Fanghui Chen, Fengwei Li, Hirpha Ketema, Yafei Cai

**Affiliations:** College of Animal Science and Technology, Nanjing Agricultural University, Nanjing 210095, China; chenxinjin@stu.njau.edu.cn (X.C.); 15122126@stu.njau.edu.cn (Y.H.); 15122212@stu.njau.edu.cn (Y.W.); 2021205001@stu.njau.edu.cn (L.S.); yanhc007@163.com (H.Y.); chenfanghui2012@163.com (F.C.); 18705077303@163.com (F.L.); hirphaketema2@gmail.com (H.K.)

**Keywords:** CDK5RAP3, NLRP3 inflammasome, liver injury, hepatic inflammation, apoptosis, pyroptosis

## Abstract

**Background/Objectives**: CDK5RAP3 (CDK5 regulatory subunit-associated protein 3), is a ubiquitously expressed protein in mammalian tissues, with emerging evidence suggesting its critical role in liver hypoplasia. CDK5RAP3 knockout results in liver hypoplasia and liver injury in mice, and most liver injuries are associated with inflammation. However, the connection between its deficiency and liver inflammation remains unclear. The NLRP3 inflammasome is a ubiquitously expressed inflammatory pathway, and growing evidence links it to liver diseases. Therefore, we aim to investigate the relationship between CDK5RAP3 deficiency in the liver and the NLRP3 inflammasome. **Methods**: To clarify the pathological link between CDK5RAP3 deficiency and liver inflammation, we developed liver-specific CDK5RAP3 knockout mouse models and mouse embryonic fibroblasts (MEFs) from conditional knockout mice. **Results**: CDK5RAP3 deficiency induces hepatic injury and inflammation in mice, with increased expression of NLRP3 inflammasome components (NLRP3, ASC, Caspase-1) and GSDMD, all of which promote pyroptosis. Notably, CDK5RAP3-deficient MEFs exhibit compromised proliferative capacity and elevated apoptotic rates. **Conclusions**: Our findings demonstrate that CDK5RAP3 is indispensable for maintaining hepatic homeostasis. Its deficiency can induce liver damage and inflammatory cell death in mice. Therefore, CDK5RAP3 may be a candidate for further investigation in inflammatory liver disease models.

## 1. Introduction

Globally, the incidence of liver diseases is increasing, with a higher risk of illness. Several factors, including viruses, alcohol, metabolites, toxins, and other pathogens, can impair liver function, cause acute or chronic injury and potentially lead to end-stage liver disease. Liver injuries form the pathological basis of liver disease. An inflammatory response accompanies most liver injuries. Although the inflammatory response is a protective mechanism of the body, persistent liver inflammation can worsen tissue damage and promote fibrosis development [1]. Therefore, further research into the molecular mechanisms of liver inflammation and the identification of key molecules as potential drug targets are very important for developing effective treatments for liver diseases.

CDK5RAP3 ( also known as LZAP or C53) was initially identified as a binding protein for cyclin-dependent kinase five activator [2,3]. CDK5RAP3 is highly conserved across mammals and is expressed in various organs, including the pancreas, brain, liver, heart, intestine, kidney, muscle, thymus, and lung [4]. It is involved in multiple cellular processes such as cell cycle regulation, cell damage response, apoptosis, invasion, migration, metastasis, cytoskeletal remodeling, and proliferation [5,6,7,8,9,10,11,12]. During early embryonic development, CDK5RAP3 plays a role in regulating the cell cycle progression, as well as the adhesion and migration of epidermal cells [13]. While its function is essential in normal cells, abnormal expression or dysfunction of CDK5RAP3 may be linked to certain diseases. Studies have shown that CDK5RAP3 plays a significant role in diseases such as hepatocellular carcinoma (HCC), gastric cancer, colon cancer, and gastric neuroendocrine carcinoma. Mak et al. found that CDK5RAP3 is highly expressed in liver cancer tissues, and in vitro experiments revealed that it promotes invasion and migration of liver cancer cells by downregulating the expression of the tumor suppressor gene p14 and activating the p21 protease [14]. Conversely, other studies reported relatively low CDK5RAP3 expression in liver cancer tissues and liver cancer cell lines (HCC), with expression levels significantly associated with prognosis. The long-term survival rate of patients with high expression was notably better than that of those with low expression [15]. The role of CDK5RAP3 in hepatocellular carcinoma (HCC) remains unclear. However, subsequent studies have established a clear link between its deficiency and liver development. Yang et al. found that CDK5RAP3 knockout causes liver hypoplasia in mice and leads to embryonic lethality [16]. CDK5RAP3 knockout mice displayed prenatal lethality with severe liver hypoplasia, as characterized by delayed proliferation and compromised differentiation [13].

Yan et al. have shown that CDK5RAP3 deficiency activates the NF-κB pathway and the NLRP3 inflammasome in mastitis infected with Streptococcus agalactiae. The role of NOD-like receptor protein 3 (NLRP3) inflammasome in alcoholic and non-alcoholic steatohepatitis, hepatitis, nanoparticle-induced liver injury, and other liver diseases has recently attracted widespread attention from clinicians and researchers [17]. The NLRP3 inflammasome is a critical pattern recognition receptor (PRR) mainly found in the cytoplasm and is a part of the NLRP protein family. It is commonly present in the cytoplasm of both immune and non-immune cells and is currently the most thoroughly studied inflammasome [18]. The NLRP3 inflammasome functions as a sensor, responding to microbial infections, endogenous danger signals, and environmental stimuli. These stimuli disrupt intracellular homeostasis, promoting the production and release of NLRP3 signal transduction mediators (such as cytokines), inducing inflammatory cell death, and, when excessively activated, contributing to the development of various inflammatory diseases [19]. Activation of the NLRP3 inflammasome occurs in two stages: the priming stage (the first signal) and the activation stage (the second signal). The priming stage increases the expression of inflammasome components NLRP3, Caspase-1, and precursor IL-1β. After activation, NLRP3 molecules oligomerize through homotypic interaction via the NACHT domain [20]. NLRP3 oligomers recruit ASC through PYD–PYD interactions, promoting the formation of ASC speckles. Subsequently, precursor Caspase-1 is recruited via the C-terminal Caspase Recruitment Domain (CARD) of ASC, and self-cleavage occurs between the p20 and p10 subunits, activating Caspase-1 [21,22]. Activated Caspase-1 precursor IL-1β and IL-18 convert into their mature forms. Additionally, activated Caspase-1 cleaves and activates gasdermin D (GSDMD), causing it to translocate to the plasma membrane where it forms pores, facilitating the release of mature IL-1β and IL-18 into the extracellular space. The massive release of inflammatory factors culminates in a programmed inflammatory form of cell death called pyroptosis [23,24,25].

In this study, liver-specific CDK5RAP3 knockout mice and a mouse embryonic fibroblast (MEF) cell line with conditional CDK5RAP3 knockdown were used for biological research. Hepatic CDK5RAP3 deficiency caused hepatic injury, increased expression of factors related to the NLRP3 inflammatory pathway, and was accompanied by Caspase-1-mediated inflammatory cell death. Meanwhile, MEFs with conditional CDK5RAP3 knockdown showed reduced growth rates, lower viability, and increased apoptosis. The levels of pro-inflammatory factors and certain genes related to the NLRP3 pathway were elevated in these MEFs, consistent with the findings in CDK5RAP3 knockout mice livers.

## 2. Materials and Methods

### 2.1. Cell Culture

Immortalized mouse embryonic fibroblasts (MEFs) were derived from *CDK5RAP3*^f/f^: CAG-CreERT2 embryos at embryonic days 13–14 (E13–E14). NCG-H11-CAG-CreERT2-polyA mice (Strain NO. T063921) were purchased from GemPharmatech (Nanjing, China). Genotypic verification confirmed homozygous floxed alleles (*CDK5RAP3*^f/f^) with either CreERT2/ERT2 or CreERT2 +/− expression. The *CDK5RAP3*^f/f^: CAG-CreERT2 mice used in this study were on a C57BL/6J background. Dissected trunk tissues were washed in phosphate-buffered saline (PBS), mechanically minced, and subjected to enzymatic digestion using 0.05% trypsin-EDTA (Gibco, New York, NY, USA) at 37 °C for 5 min. Cell suspensions were filtered through 70 μm strainers, and adherent fibroblasts were selected by differential attachment in Dulbecco’s Modified Eagle Medium (DMEM; HyClone, Logan, UT, USA) supplemented with 10% fetal bovine serum (FBS, Gibco, New York, NY, USA) and 1% penicillin-streptomycin (Solarbio, Beijing, China). At 80–90% confluence, cells were transduced with lentiviral particles carrying the T antigen (MOI = 5). Immortalization was carried out using lentiviral particles encoding SV40 large T antigen at a titer of 1 × 10^8^ TU/mL. Puromycin selection (10 μg/mL; InvioGen, San Diego, CA, USA) was initiated 48 h after transduction to enrich stably transduced cells. For conditional CDK5RAP3 knockout, MEFs were treated with 4-OHT (4-Hydroxytamoxifen, eight μM, H7904 or H6278, Sigma, St. Louis, MO, USA) for 72 h, with vehicle control (0.1% ethanol) included for normalization. All cultures were maintained at 37 °C under 5% CO_2_ atmosphere.

### 2.2. Animal Model

Liver-specific CDK5RAP3 knockout (CKO) mice, generated in Prof. Yafei Cai’s laboratory, and 5-month-old wild-type littermates were kept under controlled environmental conditions with a temperature of 18–22 °C and humidity of 50–60% in a specific pathogen-free (SPF) facility. Animals received standard laboratory chow (Research Diets, New Brunswick, NJ, USA) and autoclaved water ad libitum. All experimental animal procedures were performed in full accordance with the National Institutes of Health Guidelines for the Care and Use of Laboratory Animals.

### 2.3. Tissue Pathology Analysis

Liver tissues from wild-type and CDK5RAP3-knockout mice were fixed in 10% neutral buffered formalin (NBF; Sigma-Aldrich, St. Louis, MO, USA) for 24 h at 4 °C. Fixed tissues were dehydrated through a graded ethanol series: 75% ethanol for 30 min, 85% ethanol for 60 min, 95% ethanol for 60 min, and two changes of absolute ethanol for 1 h each to remove water. After dehydration, tissues were cleared by immersion in xylene twice for 30 min each. Infiltration was then performed with paraffin: three changes of paraffin at 60 °C for 1 h each. Tissues were embedded in fresh paraffin using embedding molds. Embedded blocks were sectioned into 5 μm-thick slices using a microtome.

Serial sections of 5 μm thickness were dewaxed by immersing in xylene three times for 20 min each. They were then rehydrated through a graded ethanol series: 100% ethanol (twice for 2 min each), followed by 95%, 90%, 80%, and 70% ethanol (2 min each). Sections were rinsed in Dulbecco’s phosphate-buffered saline (DPBS, pH 7.4; Gibco, Grand Island, NY, USA) with three 5-min washes to remove residual ethanol.

Nuclear staining was carried out by immersion in Mayer’s hematoxylin (Sigma-Aldrich, St. Louis, MO, USA) for 90 s at 25 °C. After rinsing in tap water to stop the reaction, sections were differentiated in 0.1% HCl-ethanol (*v*/*v*) for 30 s and washed in DPBS (2 × 5 min). Cytoplasmic counterstaining was performed with eosin Y (0.5% aqueous solution; Solarbio, G1120) for 120 s. Stained sections were dehydrated through an ascending ethanol series (70% → 80% → 90% → 100%; 2 min each), cleared in three changes of xylene 5 min each, and mounted with neutral balsam (Sigma-Aldrich, St. Louis, MO, USA). Slides were air-dried in a chemical fume hood for 24 h before bright-field microscopy analysis (Nikon Eclipse Ni-U, Tokyo, Japan)

### 2.4. Immunofluorescence Staining

After collection, cells were fixed with 4% paraformaldehyde (PFA) for 15 min. This was followed by permeabilization with PBST (Sigma-Aldrich, St. Louis, MO, USA) for 10 min, and subsequent incubation in blocking buffer (PBST supplemented with 10% normal goat serum; Abcam, Cambridge, MA, USA) at room temperature for 30–45 min. Primary antibodies (details in Appendix A) targeting Alb, CDK5RAP3, IL-6, and Nlrp3 were applied at 37 °C for 45–60 min. After washing with PBS containing 1% BSA, cells were incubated at 37 °C with species-specific secondary antibodies Goat Anti-Rabbit IgG H&L (Alexa Fluor^®^ 488) and Goat Anti-Mouse IgG H&L (Alexa Fluor^®^ 647); (details in Appendix A). Unbound secondary antibodies were removed with three 5-min PBS washes, and nuclei were counterstained with Hoechst 33,342 for 5 min. Fluorescent images were captured using a Zeiss LSM 900 META confocal microscope (Zeiss LSM 900 META, Jena, Germany).

### 2.5. Protein Extraction from Liver Tissues and Cells

When the cells reached approximately 80% confluence, the cell culture dishes were rinsed twice with preheated PBS (pH 7.4). Cells were lysed on ice for 30 min using RIPA buffer (Beyotime, ST506, Shanghai, China) supplemented with 1 mM phenylmethylsulfonyl fluoride (PMSF, Beyotime, ST506, Shanghai, China). Lysates were centrifuged at 12,000× *g* for 4 min at 4 °C to pellet cellular debris. The supernatant was transferred to fresh pre-chilled tubes, and protein concentration was quantified using the Pierce™ BCA Protein Assay Kit (Thermo Scientific, Shanghai, China). The protein expression levels were normalized to the housekeeping protein GAPDH. Protein denaturation was performed by adding 5 × SDS-PAGE loading buffer containing 10% β-mercaptoethanol (final concentration: 2%) to the protein lysate. Aliquots of processed protein samples were stored at −20 °C for short-term preservation.

### 2.6. RNA Extraction from Liver Tissue and Cells

Total RNA was extracted from cells at 80% confluence using TRIzol™ Reagent (Invitrogen, Carlsbad, CA, USA). Briefly, cells were washed twice with RNase-free PBS and harvested by centrifugation at 1000× *g* for 5 min at 4 °C. Cell pellets were lysed in 1 mL of TRIzol™ with vigorous pipetting on ice for 5 min. Chloroform was added, followed by vortexing for 15 s and incubation at 25 °C for 3 min. After centrifugation at 12,000× *g* for 15 min at 4 °C, the aqueous phase was transferred to fresh RNase-free tubes. RNA was precipitated by adding 0.5 mL of isopropanol and incubating at room temperature for 10 min, followed by centrifugation at 12,000× *g* for 10 min at 4 °C. The RNA pellet was washed with 1 mL of 75% ethanol (prepared in DEPC-treated water), centrifuged at 12,000× *g* for 5 min, and air-dried for 5–10 min. Purified RNA was dissolved in 30–50 μL of RNase-free water, heated at 55 °C for 10 min, and quantified using a NanoDrop™ One spectrophotometer (Shanghai Jiapeng Technology Co., Ltd., Shanghai, China), with RNA integrity confirmed by A260/A280 ratios of 1.8–2.1. RNA integrity was verified with an Agilent Bioanalyzer 2100, with all samples achieving a RIN ≥ 8.0 to ensure suitability for downstream qPCR analysis.

### 2.7. RT-qPCR

According to the manufacturer’s instructions (TransGen Biotech, AT311, Beijing, China), RNA was reverse transcribed into cDNA using a reverse transcription system. The components included 1 µL of RNA, 1 µL of Anchored Oligo (dT)18 Primer (0.5 µg/µL), 10 µL 2 *TS Reaction Mix, 1 µL of TransScript^®^ RT/RI Enzyme Mix, 1 µL of gDNA Remover, and 6 µL RNase-free Water (TransGen Biotech, Beijing, China). The mixture was gently mixed, incubated at 42 °C for 15 min, and then heated at 85 °C for 5 s. Real-time quantitative PCR was performed on the instrument using appropriate primers. The primer sequences of target genes and the GAPDH gene were designed using Primer Premier 5.0 software.

### 2.8. Western Blot Analysis

Electrophoresis was conducted at 90 V for approximately 30 min, followed by 120 V for 1 h. After electrophoresis of the protein solution, proteins were transferred onto a PVDF membrane. Membrane transfer was performed at a constant current of 200 mA with electroblotting on ice for 100 min. Membranes were blocked with 5% skimmed milk powder in TBST for 1 h before incubation with primary antibodies overnight at 4 °C. The antibodies used in this study included: GAPDH, CDK5RAP3, Bax, Bcl2, TNFα, cleaved-IL-1β, IL-1β, IL-6, NLRP3, ASC, caspase1, cleaved-caspase1, and GSDMD. Subsequently, the secondary antibody was incubated at room temperature for 1 h. Finally, the membranes were processed using an enhanced chemiluminescence detection system. Densitometry was performed using ImageJ 1.54g (National Institutes of Health, Bethesda, MD, USA).

### 2.9. Other Related Operations

Crystal violet (Macklin 548-62-9, Shanghai, China) was used for staining with crystal violet. The Annexin V-FITC/PI cell apoptosis detection kit (Vazyme, A211, Nanjing, China) was used according to the manufacturer’s instructions. More details can be found in Appendix A and Methods (Appendix A).

### 2.10. Statistical Analysis

Data analysis was performed using GraphPad Prism 8 software (GraphPad Software, San Diego, CA, USA). An unpaired *t*-test was used to compare two groups. Data were expressed as mean ± SEM and were presented as three independent experiments (*n* = 3 per group). * *p* < 0.05, ** *p* < 0.01, *** *p* < 0.001, and **** *p* < 0.0001.

## 3. Results

### 3.1. CDK5RAP3 Deletion Induced Liver Injury in Mice

*CDK5RAP3*^f/f^ mice were crossed with Alb-Cre mice to generate hepatocyte-specific CDK5RAP3 knockout mice (*CDK5RAP3*^Δ/Δhep^). Western blot and quantitative PCR (qPCR) analysis showed that the levels of CDK5RAP3 protein and mRNA in the liver of *CDK5RAP3*^Δ/Δhep^ mice were significantly reduced compared to those of *CDK5RAP3*^f/f^ mice (Figure 1A). Immunofluorescence (IF) staining revealed similar albumin (Alb) signal intensity and distribution in both *CDK5RAP3*^f/f^ and *CDK5RAP3*^Δ/Δhep^ groups, indicating preserved basal hepatocyte function and excluding widespread cellular dysfunction. Notably, a reduced CDK5RAP3 signal was specifically observed in Alb^+^ hepatocyte regions, while stromal cells (Alb^−^ regions) maintained basal CDK5RAP3 expression (Figure 1B). Hematoxylin-eosin (H&E) staining of liver sections showed that, compared with the liver tissue of *CDK5RAP3*^f/f^ control mice, the liver tissue of *CDK5RAP3*^Δ/Δhep^ mice exhibited morphological changes (Figure 1C), characterized by pathological features such as lipid accumulation, vacuolization, and inflammatory cell infiltration in the hepatic parenchyma. These changes persisted in the *CDK5RAP3*^Δ/Δhep^ group, while they were rare or absent in *CDK5RAP3*^f/f^ liver tissue. Furthermore, in the *CDK5RAP3*^Δ/Δhep^ group, protein levels of the pro-apoptotic factor BAX increased, mRNA expression was significantly higher, and the BAX/BCL2 ratio was elevated (Figure 1D), indicating that cellular apoptosis occurred in liver tissue. These findings suggested that the absence of CDK5RAP3 causes liver dysfunction and hepatocyte injury.

### 3.2. CDK5RAP3 Deletion Is Associated with Increased Liver Pro-Inflammatory Factor Expression

After observing inflammatory signs in H&E staining liver tissue, liver pro-inflammatory factors were systematically quantified to validate the inflammatory phenotype. Western blot analysis showed that the protein levels of IL-6 and mature IL-1β were higher in the *CDK5RAP3*^Δ/Δhep^ group compared to the *CDK5RAP3*^f/f^ group (Figure 2A). qPCR further confirmed that the mRNA expression levels of TNF-α, IL-6, and IL-1β were increased in the *CDK5RAP3*^Δ/Δhep^ group (Figure 2B). These results are consistent with the increased IL-6 signal observed in immunofluorescence staining (Figure 3C) and the histopathological evidence of inflammation seen in liver sections, indicating that an association between CDK5RAP3 gene deletion and pro-inflammatory cytokine dysregulation was confirmed.

### 3.3. CDK5RAP3 Deficiency Is Associated with Increased Expression of NLRP3 Pathway Components, and Is Accompanied by Pyroptosis-Associated Molecular Alterations

To investigate the role of CDK5RAP3 in inflammatory signaling pathways, we systematically evaluated the expression of key components of the NLRP3 inflammasome. Western blot and qPCR analyses demonstrated that both protein levels and mRNA expression of NLRP3 and ASC were significantly elevated in the livers of *CDK5RAP3*^Δ/Δhep^ mice compared to controls (Figure 3A,B). Concurrently, total Caspase-1 protein showed a non-significant increasing trend, while its mRNA expression was significantly upregulated. GSDMD mRNA levels were also markedly increased (Figure 3A,B). Immunofluorescence staining showed increased IL-6 and NLRP3 signals along with the formation of punctate co-localization structures (Figure 3C). Furthermore, Western blot analysis detected accumulation of mature IL-1β (p17) (Figure 2A), indicating enhanced processing of this cytokine. Collectively, these findings demonstrate that CDK5RAP3 deficiency upregulates NLRP3 inflammasome components, enhances IL-1β maturation, and is accompanied by molecular changes associated with pyroptosis. These molecular changes suggest a potential link to pyroptotic cell death.

### 3.4. Loss of CDK5RAP3 Is Associated with Apoptosis and Inflammatory Responses

To investigate the regulatory mechanism of CDK5RAP3 deficiency in apoptotic pathways, a conditional knockout model of mouse embryonic fibroblasts (MEFs) was established using an inducible system with 4-hydroxytamoxifen (4-OHT, eight μM). After 48 h of 4-OHT treatment, CDK5RAP3 protein expression in MEFs was reduced (Figure 4A). Crystal violet staining (24–96 h) showed that t cell proliferation in the 4-OHT-treated group was visibly slower compared to the EtOH control group (Figure 4B,C). Flow cytometry (with Annexin V/PI double staining) demonstrated an increased proportion of apoptotic cells in the 4-OHT-treated group (EtOH group: 4.63%; 4-OHT group: 5.83%, 1.262-fold increase, Figure 4D), along with an elevated BAX/BCL2 ratio (Figure 5A). Consistent with the liver experimental results, 4-OHT treatment upregulated the protein levels of the NLRP3 inflammasome and its downstream factor GSDMD (Figure 5B), with simultaneous increases in the expression of pro-inflammatory cytokines IL-6 and IL-1β (Figure 5C). Overall, data from cell trauma and MEFs suggest that deletion of CDK5RAP3 may exacerbate liver injury by synergistically enhancing inflammatory and apoptotic pathways.

## 4. Discussion

This study used liver-specific *CDK5RAP3*^Δ/Δhep^ mice models and conditional knockout MEF cell models to demonstrate how CDK5RAP3 deficiency worsens liver injury and inflammasome by activating NLRP3 inflammasome-mediated crosstalk between pyroptosis and apoptosis.

Preliminary histological analysis using H&E staining showed that *CDK5RAP3*^Δ/Δhep^ mice displayed significant phenotypic abnormalities, including hepatocellular swelling, vacuolization, lipid deposition, and infiltration of inflammatory cells. Subsequent Western blot and quantitative PCR confirmed that CDK5RAP3 was significantly downregulated at both protein and mRNA levels in the liver tissues of these mice. Immunofluorescence staining demonstrated weakened co-localization signals between albumin (Alb) and CDK5RAP3, indicating impaired functional integrity of hepatocytes. Crystal violet staining of 4-hydroxytamoxifen (4-OHT)-induced conditional knockout (CKO) mouse embryonic fibroblasts (MEFs) revealed a reduction in cell numbers compared to wild-type controls, further confirming decreased proliferative capacity. These phenotypes were closely associated with abnormal activation of apoptotic pathways. Subsequent studies found increased levels of the pro-apoptotic factor BAX in *CDK5RAP3*^Δ/Δhep^ mice, leading to a significant rise in the BAX/Bcl-2 ratio. Annexin V/PI double-labeled flow cytometry confirmed an increase in apoptotic cells, aligning with the elevated BAX/Bcl-2 ratio. These results highlight the critical role of CDK5RAP3 in maintaining liver function and hepatocellular homeostasis, consistent with Yang et al.’s findings that CDK5RAP3 deficiency causes embryonic liver developmental defects [13].

Besides obvious phenotypic abnormalities in the liver, CDK5RAP3 deficiency in mice showed significant increases in pro-inflammatory cytokines (e.g., IL-6, IL-1β, TNF-α) at both protein and mRNA levels. This phenotype is closely associated with abnormal activation of the NLRP3 inflammasome, a key driver of hepatitis that initiates inflammatory cascades through Caspase-1-mediated maturation of IL-1β and IL-18 [26,27]. Our data not only showed upregulated expression of NLRP3 and Caspase-1 in the livers of *CDK5RAP3*^Δ/Δhep^ mice but also revealed significant increases in ASC and GSDMD at both mRNA and protein levels. These results are consistent with studies by Mridha et al., highlighting the central role of the NLRP3 inflammasome in liver injury pathologies such as liver fibrosis and steatosis [28,29]. Immunofluorescent labeling further localized increased NLRP3 and ASC speck formation in CDK5RAP3-deficient hepatocytes, confirming inflammasome assembly at the cellular level and suggesting that CDK5RAP3 may participate in liver homeostasis maintenance by regulating NLRP3 inflammasome assembly or activity.

Existing studies have shown that NLRP3 inflammasome activation depends on “dual-signal” regulation: the first signal induces transcription of NLRP3 and pro-IL-1β through the NF-κB pathway, while the second signal triggers inflammasome assembly via potassium efflux, mitochondrial damage, and other mechanisms [19,22]. In this study, CDK5RAP3 deficiency simultaneously upregulated NF-κB downstream target genes (e.g., IL-6, TNF-α) and NLRP3 inflammasome components, suggesting it may regulate the process by affecting the NF-κB pathway or directly interacting with inflammasome components. This study only provides relevant evidence, and the mechanistic relationships remain to be established. For instance, previous research has found that certain proteins can inhibit NLRP3 oligomerization by binding to its LRR domain [30]; whether CDK5RAP3 inhibits NLRP3 activation through similar mechanisms requires further validation using co-immunoprecipitation or proximity ligation assays.

This study found that CDK5RAP3 deficiency simultaneously activates apoptotic (BAX upregulation) and pyroptotic (GSDMD cleavage) pathways. This phenomenon may involve mutually reinforcing molecular mechanisms: on one hand, Caspase-1-mediated GSDMD cleavage induces pyroptosis, releasing damage-associated molecular patterns (DAMPs) such as ATP and HMGB1 [31]; on the other hand, these DAMPs can serve as secondary signals for NLRP3 inflammasome activation, further exacerbating inflammation and apoptosis [32]. Although this study did not directly detect DAMP release, existing research has confirmed that GSDMD-mediated pyroptosis can amplify NLRP3 inflammasome activation through the ATP-P2 × 7 receptor axis [25,33], providing a theoretical basis for explaining the “inflammation-death” positive feedback observed in this study.

To verify CDK5RAP3 deficiency’s causal role in inflammation-driven apoptosis, we developed a tamoxifen-inducible knockout model using mouse embryonic fibroblasts (MEFs). The experimental MEFs were treated with 4-OHT, while controls received ethanol (EtOH). Treatment with 4-OHT significantly increased the apoptosis rate, inhibited cell growth, and decreased CDK5RAP3 expression levels. In addition, there was an increase in inflammatory cytokines and NLRP3 pathway components. WB and q-PCR analyses confirmed a significant increase in the Bax/Bcl-2 ratio. These results indicate that CDK5RAP3 deficiency initiates a self-amplifying “inflammation-death” cycle, exacerbating embryonic fibroblasts’ damage through synergistic of apoptosis and pyroptosis. This study used MEF cells to validate CDK5RAP3 function, revealing similar inflammatory and apoptotic phenotypes to hepatocytes; however, physiological microenvironments differ between fibroblasts and hepatocytes (e.g., metabolic characteristics, immune cell interactions). Nevertheless, existing studies have shown that activation of hepatic stellate cells (a type of liver parenchymal fibroblast) plays a key role in liver inflammation and fibrosis [1], and MEF cell models are commonly used to validate conserved mechanisms of inflammation-related pathways [34]. Combined with in vivo mouse experimental results, MEF data still provide important support for CDK5RAP3 function, though future validation using primary hepatocytes or organoid models is warranted.

This study reveals the protective role of CDK5RAP3 in hepatic inflammation, providing a new research perspective for the treatment of chronic liver disease. Compared with existing NLRP3 inhibitors such as MCC950, which primarily interacts directly with the Walker B motif within the NLRP3 NACHT domain to block ATP hydrolysis [35], CDK5RAP3 may work by simultaneously regulating inflammasome activation and apoptotic pathways (Figure 6). Its high level expression in liver tissue [13] also suggests potential for organ-specific intervention. However, it is important to note that, due to the current incomplete understanding of its mechanism, including unclear interactions with pathways like NF-κB or MAPK [36,37], future research should focus on clarifying these interaction mechanisms and verifying whether they directly bind to ASC or NLRP3 to regulate inflammasome assembly through co-immunoprecipitation experiments, which would set the stage for assessing their clinical potential.

Future research may focus on three areas: (1) elucidating the interaction between CDK5RAP3 and NLRP3 inflammasome components through structural biology methods; (2) validating the regulatory role of CDK5RAP3 in disease models such as NASH and liver fibrosis; and (3) exploring screening strategies for CDK5RAP3 activators to identify candidate molecules for clinical use.

Despite the valuable insights gained from this study, several limitations should be acknowledged. First, no functional inflammasome activation assays (e.g., caspase-1 activity measurement or IL-1β release assays under stimulated conditions) were performed to directly confirm NLRP3 inflammasome activation, which limits the mechanistic depth of the observed associations. Second, the use of mouse embryonic fibroblasts (MEFs) as a cellular model may not fully recapitulate the hepatocyte-specific microenvironment, potentially restricting the generalizability of conclusions to liver parenchymal cells. Third, the sample size in animal and cell experiments was relatively small (n = 3 per group), which may affect the statistical robustness of the findings. Finally, all observations are correlative in nature, and causal relationships between CDK5RAP3 deficiency, NLRP3 inflammasome activation, and hepatic inflammation require further validation through interventional studies. These limitations highlight the need for cautious interpretation of the results and underscore directions for future research to address these constraints.

## Figures and Tables

**Figure 1 biomedicines-13-02030-f001:**
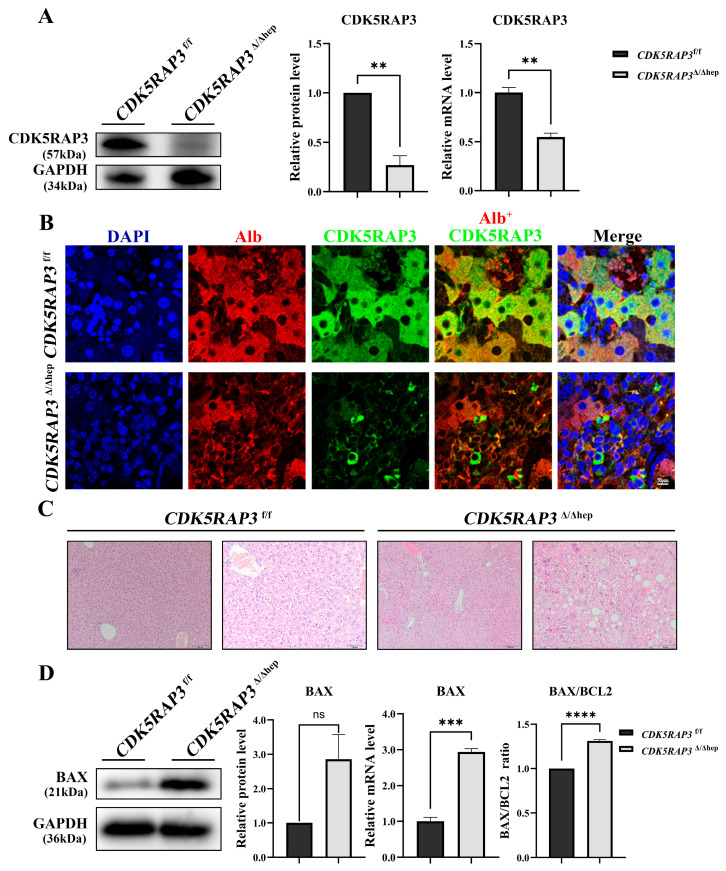
Deletion of CDK5RAP3 induced liver injury. (**A**) Analysis of CDK5RAP3 (57 kDa) protein and mRNA levels in the liver of *CDK5RAP3*^f/f^ and *CDK5RAP3*^Δ/Δhep^ mice. (**B**) Immunofluorescence (IF) staining of liver sections from *CDK5RAP3*^f/f^ and *CDK5RAP3*^Δ/Δhep^ mice for albumin (Alb, red) and CDK5RAP3 (green). Nuclei were counterstained with DAPI (blue). Representative images are shown (scale = 50 μm). (**C**) Histochemical analysis of liver tissues from *CDK5RAP3*^Δ/Δhep^ and WT mice by HE analysis (scale = 100 μm/50 μm). (**D**) Protein and mRNA expression of BAX (21 kDa) in extracted liver tissues, along with the calculation of the BAX/BCL2 expression ratio in *CDK5RAP3*^f/f^ and *CDK5RAP3*^Δ/Δhep^ mice. Data were expressed as mean ± standard error (n = 3). ** *p* < 0.01; *** *p* < 0.001; **** *p* < 0.0001; ns = not significant.

**Figure 2 biomedicines-13-02030-f002:**
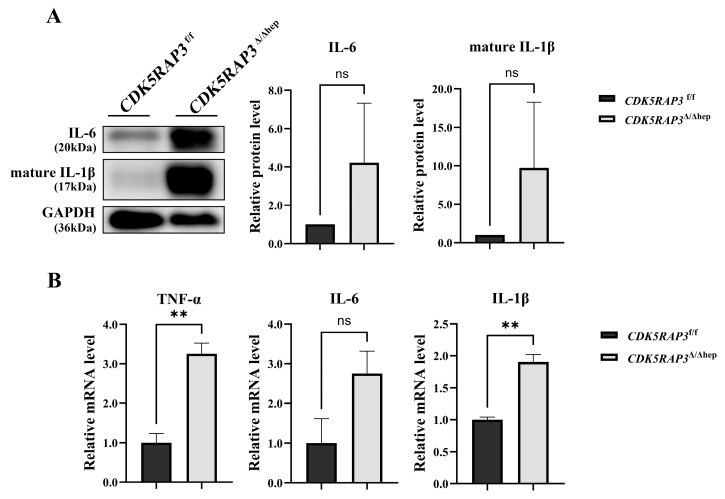
CDK5RAP3 deletion upregulates pro-inflammatory cytokine expression. (**A**) Western blot analysis of IL-6 (20 kDa) and mature IL-1β (17 kDa) protein levels in liver tissues from *CDK5RAP3*^f/f^ and *CDK5RAP3*^Δ/Δhep^ mice. (**B**) mRNA expression of TNF-α, IL-6, and IL-1β in liver tissues extracted from *CDK5RAP3*^f/f^ and *CDK5RAP3*^Δ/Δhep^ mice. Data are shown as mean ± standard error (n = 3). ** *p* < 0.01; ns = not significant.

**Figure 3 biomedicines-13-02030-f003:**
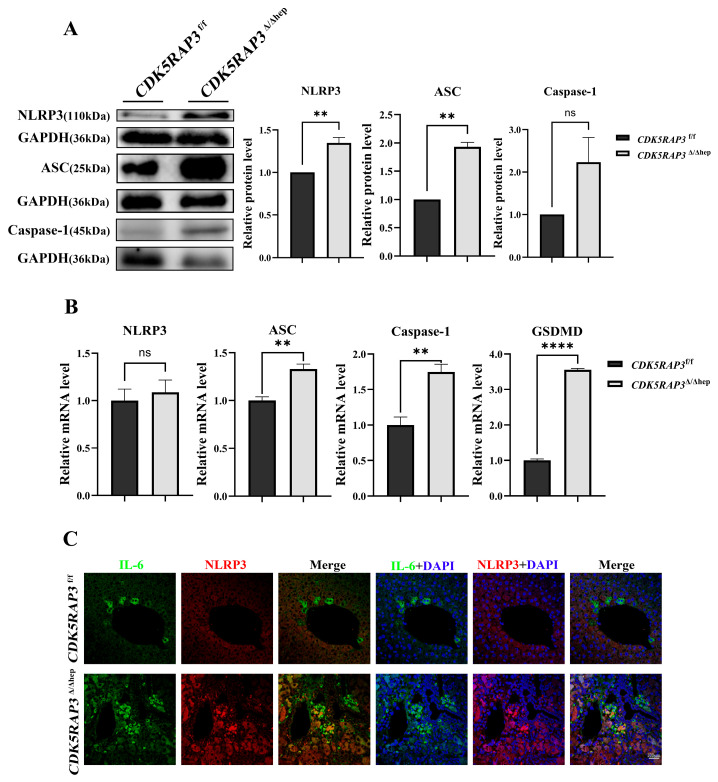
CDK5RAP3 deficiency is associated with increased NLRP3 pathway component expression, increases IL-1β maturation, and is accompanied by pyroptosis-associated molecular alterations. (**A**) Western blot analysis of NLRP3 (110 kDa), ASC (25 kDa), and Caspase-1 (45 kDa) protein levels in liver tissues extracted from *CDK5RAP3*^f/f^ and *CDK5RAP3*^Δ/Δhep^ mice. (**B**) mRNA expression of NLRP3, ASC, Caspase-1, and GSDMD in liver tissues from the same groups. (**C**) Liver tissues from *CDK5RAP3*^f/f^ and *CDK5RAP3*^Δ/Δhep^ mice were subjected to immunofluorescence staining to detect IL-6 and NLRP3 (scale bar = 100 μm). Images shown for qualitative comparison only. Data are expressed as mean ± standard error (n = 3). ** *p* < 0.01; **** *p* < 0.0001; ns = no significant. Data show protein/mRNA levels only; functional activation not assessed.

**Figure 4 biomedicines-13-02030-f004:**
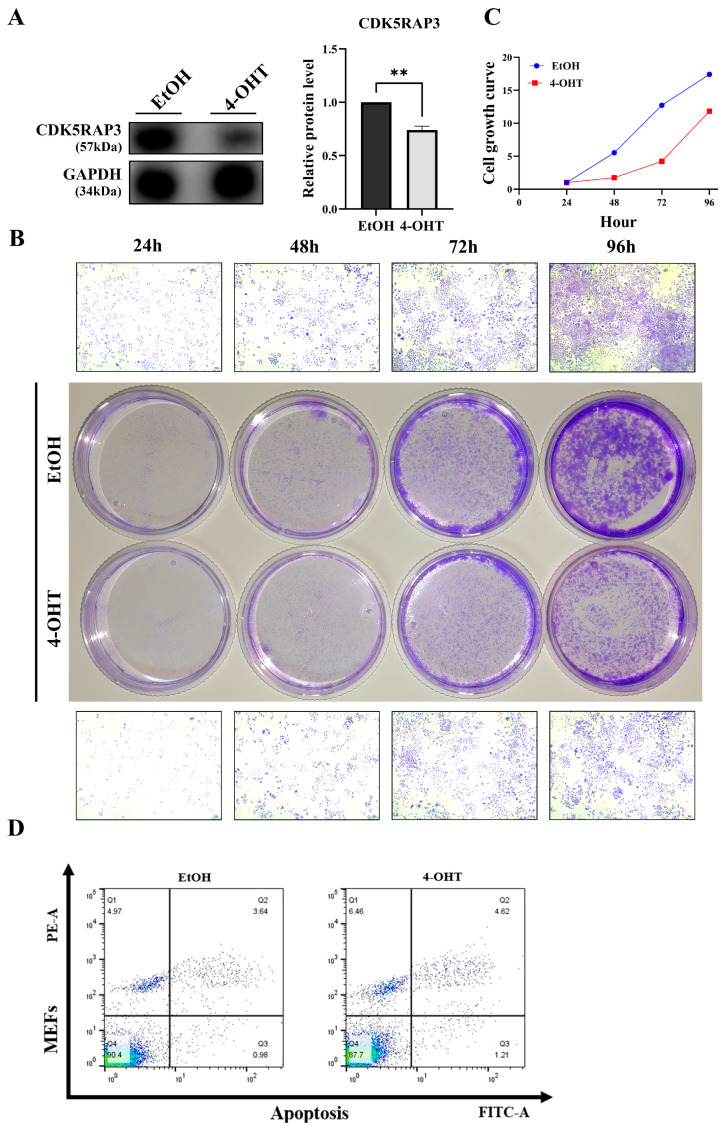
CDK5RAP3 deletion correlates with apoptosis and inflammatory feedback. (**A**) Western blot validation of CDK5RAP3 (57 kDa) protein expression in MEFs after 48-h treatment with 4-OHT (8 μM). (**B**) Cell growth curves of MEFs treated with EtOH or 4-OHT (8 μM) for 24–96 h. (**C**) Representative crystal violet staining images of MEFs treated with EtOH or 4-OHT (8 μM) at indicated time points (scale bar = 101 μm). (**D**) Apoptotic cell percentages detected by Annexin V/PI double staining flow cytometry. Data are expressed as mean ± standard error (n = 3). ** *p* < 0.01.

**Figure 5 biomedicines-13-02030-f005:**
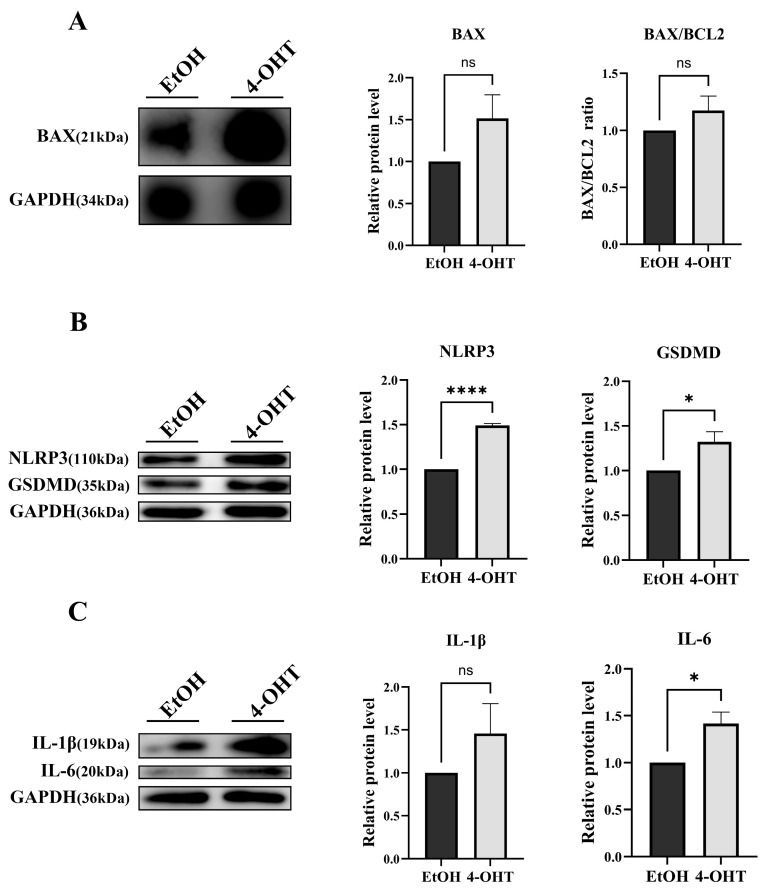
Deficiency of CDK5RAP3 is associated with NLRP3 inflammasome signaling in MEFs, accompanied by the release of inflammatory cytokines. (**A**) Western blot analysis of BAX (21 kDa) protein expression and calculation of the BAX/BCL2 ratio in MEFs treated with 4-OHT (8 μM). (**B**) Western blot analysis of NLRP3 (110 kDa) and GSDMD (35 kDa) protein expression in MEFs treated with EtOH (8 μM) and 4-OHT (8 μM). (**C**) Western blot analysis of IL-6 (20 kDa) and IL-1β (19 kDa) protein expression in MEFs treated with EtOH (8 μM) and 4-OHT (8 μM). Data are expressed as mean ± standard error (n = 3). * *p* < 0.05; **** *p* < 0.0001; ns = no significant.

**Figure 6 biomedicines-13-02030-f006:**
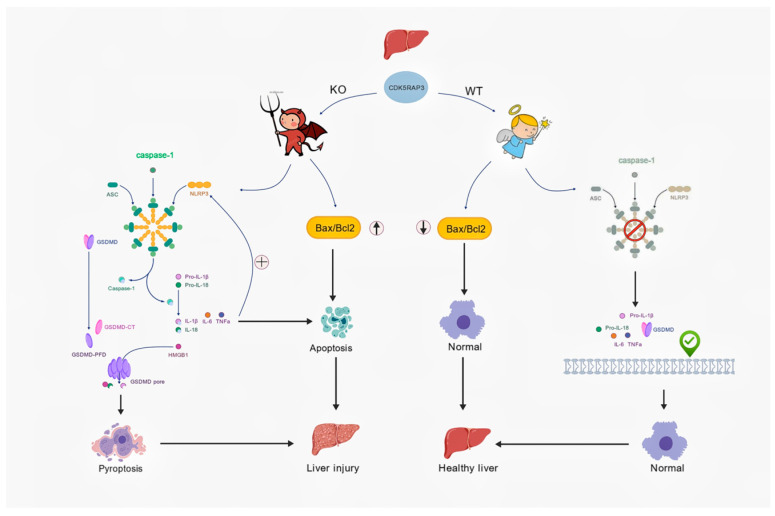
CDK5RAP3 deficiency exacerbates hepatic inflammation via NLRP3 inflammasome-mediated pyroptosis and apoptotic crosstalk. A schematic illustration shows the pathological cascade caused by CDK5RAP3 deficiency in mouse livers, involving of inflammasome-mediated pyroptosis and apoptosis.

## Data Availability

The raw data supporting the conclusions of this study are available from the corresponding author upon reasonable request. Interested researchers may contact Yafei Cai (ycai@njau.edu.cn) for data access.

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
