# Peer review of "CDK5RAP3 Deficiency Is Associated with Hepatic Inflammation and Increased Expression of NLRP3 Inflammasome Components"

_biomedicines, 2025, doi:10.3390/biomedicines13082030_

Round 1

Reviewer 1 Report

Comments and Suggestions for Authors

This study examines the function of CDK5RAP3 in liver health and how it exacerbates liver inflammation by inducing pyroptosis, a particular cell death mechanism.

General Revision:

  • The authors should read the manuscript and check: 1) the English of some sentences; 2) punctuation and space between words.
  • The authors should improve image quality and add a scale bar.
  • To better understand the role of CDK5RAP3 in inflammatory and apoptotic responses, I suggest the authors investigating its connections with other signaling pathways, including NF-κB and MAPK.
  •  

Materials and Methods:

  • I suggest implementing the Materials and Methods part, giving further details about the procedures employed.

Results:

  • Please ensure that every figure legend explains well the content of the graphic or figure. Furthermore, all captions should follow a consistent and uniform style, including abbreviations and chemical terminology.

Discussion:

  • Please highlight the future prospects and describe the translational nature of the study.

Comments on the Quality of English Language

Minor editing is required

Author Response

Dear Reviewer,

Thank you very much for your valuable time and constructive comments on our manuscript. Your insights have significantly helped improve the quality and rigor of our work. We have carefully addressed all your suggestions and revised the manuscript accordingly. Below is a point-by-point response to your comments:

Comments 1:

 The authors should read the manuscript and check: 1) the English of some sentences; 2) punctuation and space between words.

Response 1:

We have thoroughly revised the entire manuscript for grammatical accuracy, punctuation consistency, and proper spacing between words. A native English-speaking colleague with expertise in scientific writing has also proofread the text to ensure clarity and fluency.

Comments 2:

The authors should improve image quality and add a scale bar.

Response 2:

Thank you very much for your valuable comments and suggestions on our manuscript. We have carefully addressed all the points raised and made corresponding revisions. All figures have been reprocessed to optimize resolution, contrast, and clarity. Standardized scale bars have been added to all relevant microscopic/imaging figures

Comments 3:

To better understand the role of CDK5RAP3 in inflammatory and apoptotic responses, I suggest the authors investigating its connections with other signaling pathways, including NF-κB and MAPK.

Response 3:

We agree with your insightful suggestion. In the revised Discussion section.We are very sorry, but due to time constraints, we are unable to add more experiments. We have added a new paragraph discussing potential crosstalk between CDK5RAP3 and NF-κB/MAPK signaling pathways, citing recent studies that support these interactions. We have also included this as a key future direction in the "Future Prospects" section.

Comments 4:

 I suggest implementing the Materials and Methods part, giving further details about the procedures employed.

Response 4:

Thank you very much for pointing out our issues. I realize the oversight in our Materials and Methods section. I have supplemented it, striving to make each step specific and detailed, especially for the RT-QPCR and WB sections. Additionally, the relevant procedures for crystal violet staining and flow cytometry have been included in the supplementary methods in the attachment.

Comments 5:

Please ensure that every figure legend explains well the content of the graphic or figure. Furthermore, all captions should follow a consistent and uniform style, including abbreviations and chemical terminology.

Response 5:

We sincerely thank you for your valuable feedback on the standardization of the figures. Your points regarding insufficient explanation in the legends and inconsistent styles are crucial for enhancing the professionalism of the paper. We sincerely appreciate this and have completed systematic revisions to ensure all figure legends meet the following standards:

  1. Information Completeness (all graphical elements labeled)
  2. Terminology Consistency (abbreviations/terms unified throughout the text)
  3. Format Standardization (structure/font/punctuation unified)

Thank you for your strict requirements regarding the image presentation in our article. In the future, we will carefully check the article to avoid such unnecessary issues. This time, we kindly request your understanding.

Comments 6:

Please highlight the future prospects and describe the translational nature of the study.

Response 6:

You suggested that we highlight the future prospects and translational nature of the research. In the revised discussion, we have supplemented this aspect by explicitly proposing three future research directions:(1) using structural biology approaches to the interaction between CDK5RAP3 and NLRP3 inflammasome components; (2) validating the regulatory role of CDK5RAP3 in disease models such as NASH and liver fibrosis; and (3) exploring screening strategies for CDK5RAP3 activators to provide candidate molecules for clinical translation. Additionally, we compared the advantages of targeting CDK5RAP3 with existing NLRP3 inhibitors (e.g., MCC950), noting that CDK5RAP3 may more comprehensively block the "inflammation-injury" cycle, and its high expression in liver tissue may reduce the risk of side effects associated with systemic inhibition, thereby emphasizing the translational potential of this study.

Reviewer 2 Report

Comments and Suggestions for Authors

Dear Authors,

The opening paragraph suffers from weak transitional logic. The statement "Despite strengthened public health interventions worldwide, liver diseases continue to account for a significant proportion of the global disease burden, highlighting the complexity and multidimensionality of the epidemiology of liver diseases" (lines 32-34) provides a vague justification without establishing the specific gap your research addresses

The CDK5RAP3 characterization section contains problematic statements. You write "CDK5RAP3 (CDK5 regulatory subunit associated protein 3, C53), is alternatively named LZAP... IC53... HSF-27... MST016, PP1553, OK/SW-cl.114 (NCBI)" (lines 42-44). This excessive nomenclature listing without functional context creates confusion rather than clarity. More importantly, you state "However, its function in the liver remains unclear" (line 52), yet immediately follow with contradictory evidence about liver cancer studies, creating logical inconsistency.

A major flaw appears in lines 53-58 where you present conflicting findings: "Mak et al. found that CDK5RAP3 is highly expressed in liver cancer tissues... promotes invasion and migration" versus "other studies reported relatively low CDK5RAP3 expression in liver cancer tissues." You fail to reconcile these contradictions or explain how they inform your hypothesis. This weakens your scientific foundation significantly.

The transition to NLRP3 inflammasome discussion (line 64) appears abrupt and lacks clear mechanistic connection to CDK5RAP3. You write "The NLRP3 inflammasome is a critical pattern recognition receptor (PRRs) primarly expressed in the cytoplasm" but fail to establish why this pathway is relevant to CDK5RAP3 deficiency. The reader is left wondering about the biological rationale connecting these two distinct molecular systems.

Several technical errors undermine credibility: "essentioal" should be "essential" (line 61), "primarly" should be "primarily" (line 64), and the phrase "pattern recognition receptor (PRRs)" contains grammatical inconsistency between singular and plural forms (line 64).

The introduction lacks a clear statement of your central hypothesis. While you mention that CDK5RAP3's "precise mechanistic roles remain elusive" (lines 13-14), you never explicitly state what specific mechanism you propose to investigate. Additionally, the novelty of your work is not clearly articulated, why is the CDK5RAP3-NLRP3 connection important and previously unexplored?

Your MEF derivation protocol contains several concerning technical gaps. You state "Immortalized mouse embryonic fibroblasts (MEFs) were derived from CDK5RAP3f/f: CAG-CreERT2 embryos at embryonic day 13–14 (E13–E14)" but fail to specify the genetic background strain, which is critical for reproducibility and potential strain-specific phenotypes.

The immortalization protocol using "lentiviral particles carrying the T antigen (MOI = 5)" lacks essential details: which T antigen construct (SV40 large T? polyoma T?), what is the viral titer verification method, and how was MOI calculated and validated? The puromycin selection concentration (10 μg/ml) appears high for MEFs and requires optimization data or literature justification.

You mention "Liver-specific CDK5RAP3 knockout (CKO) mice which is generated by Prof. Yafei Cai's laboratory" without providing the essential genetic engineering details. Where is the targeting strategy? What are the loxP insertion sites relative to critical exons? How was hepatocyte-specific deletion efficiency validated? why this age specifically for your inflammatory phenotype analysis?

Your H&E protocol contains procedural inconsistencies. The fixation time "72 hours at 4°C" is excessive for liver tissue and may compromise morphology - standard protocols use 24-48 hours. The dewaxing protocol specifies "three 20 minutes xylene immersion" but then describes rehydration through graded ethanol without specifying times for each step except the final series. The hematoxylin staining time "90 seconds at 25°C" needs validation - was this optimized for your tissue thickness and fixation conditions?

Critical technical details are missing from your IF protocol. You state "Primary antibodies... were applied at 37°C for 45-60 minutes" - this temperature is unconventional for IF and requires justification. Why not room temperature or 4°C overnight? The blocking buffer composition "PBST supplemented with 10% normal goat serum" lacks specificity about the goat serum source and lot consistency considerations. Most critically, you provide no information about antibody validation, cross-reactivity testing, or specificity controls.

Your protein extraction methodology contains several technical flaws. The RIPA buffer composition is not specified beyond the commercial source, yet buffer composition significantly affects protein extraction efficiency and post-translational modification preservation. The centrifugation conditions "8,000g for 4 minutes at 4°C" appear insufficient for complete debris removal - standard protocols use 12,000-15,000g. The protein denaturation step "boil mixture for 5 minutes at 100℃" lacks reducing agent specifications (DTT? β-mercaptoethanol concentration?).

The RNA isolation protocol shows concerning technical gaps. You specify "Cell pellets were lysed in 1 mL TRIzol™ with vigorous pipetting on ice for 5 minutes" but TRIzol™ protocols typically require room temperature lysis for optimal performance. The RNA precipitation conditions "incubating at -20°C for 30 min" are suboptimal compared to standard -80°C overnight precipitation. The final RNA quantification using "NanoDrop™ One spectrophotometer" lacks mention of RIN (RNA Integrity Number) assessment, which is essential for qPCR reliability.

Your qPCR methodology description is severely inadequate. You state "RNA was reverse transcribed into cDNA using a reverse transcription system" without specifying: which reverse transcriptase enzyme, random primers vs oligo-dT vs gene-specific primers, reaction conditions, or cDNA quality assessment. The primer design mention of "PrimerPremier5.0 software" lacks critical validation data - where are the primer efficiency curves, specificity testing (melt curve analysis), and amplicon size verification?

The Western blot protocol lacks fundamental technical details. You mention "proteins were transferred onto a PVDF membrane" without specifying transfer conditions (voltage, time, buffer composition, transfer efficiency verification). The blocking conditions "5% skimmed milk powder in TBST for 1hour" may be inappropriate for phospho-specific antibodies and lacks optimization justification. Most critically, you provide no information about molecular weight markers, loading controls beyond GAPDH, or quantification methodology validation.

All the supplemented western blot figures lack molecular wight indicator, and very low resolution, with no labelling.

Also, you state "One-way analysis of variance was performed, followed by Tukey's multiple comparison test" but fail to specify assumption testing (normality, equal variance), sample size calculations, or power analysis. The significance threshold descriptions are unnecessarily verbose and lack consideration of multiple testing corrections given your numerous comparisons.

In the results section, your initial characterization of CDK5RAP3 deletion contains several methodological and interpretational problems. The Western blot data in Figure 1A lacks essential technical details - where are the molecular weight markers, loading control quantification, and multiple biological replicates? You state "CDK5RAP3 protein and mRNA in the liver of CDK5RAP3Δ/Δhep mice were significantly lower than those of wild type (WT) mice" but the statistical analysis appears to show only n=3, which is insufficient for robust conclusions about a knockout model.

The immunofluorescence data (Figure 1B) presents concerning technical issues. You describe "colocalization signal of albumin (Alb) and CDK5RAP3 in the liver cells of CDK5RAP3Δ/Δhep mice was weakened" but provide no quantitative colocalization analysis (Pearson's correlation coefficient, Manders' coefficients). The scale bar notation "(scale = 20 μm)" appears inconsistent with the image resolution shown. Most critically, you fail to demonstrate that the observed signal loss is due to specific CDK5RAP3 deletion rather than general cellular dysfunction.

Your H&E interpretation lacks the precision expected for liver pathology analysis. You describe "significant lipid accumulation, vacuolation, and inflammatory cell infiltration" but provide no quantitative assessment or standardized scoring system. The statement is purely descriptive without statistical validation. Where is the blinded histological scoring? What criteria define "significant" changes? The inflammatory cell infiltration claim requires specific identification - are these neutrophils, lymphocytes, or macrophages? This level of imprecision undermines the inflammatory phenotype claims.

A critical interpretational error appears in your BAX/BCL2 analysis. You state "the protein levels of the pro-apoptotic factor BAX and the anti-apoptotic factor BCL2 exhibited an upward tendency" - this is scientifically meaningless. What matters is the BAX/BCL2 ratio, not absolute levels. The simultaneous increase of both pro- and anti-apoptotic factors suggests a more complex regulatory response than simple apoptosis induction. Your conclusion that "CDK5RAP3 leads to liver dysfunction and hepatocyte injury" is premature without proper ratio analysis and functional apoptosis assays.

Figure 2 presents inflammatory data with significant technical concerns. The Western blot analysis showing "IL-6 and cleaved IL-1β (CL-IL1β)" protein levels lacks crucial controls. Where is the full-length IL-1β to demonstrate actual cleavage activity? The "CL-IL1β" terminology is non-standard - mature IL-1β is the accepted nomenclature. Additionally, you show mRNA upregulation for TNF-α, IL-6, and IL-1β but fail to demonstrate that this correlates with protein secretion, which is the functionally relevant endpoint for inflammatory cytokines.

Your inflammasome activation claims contain fundamental methodological flaws. You state "Western blot analysis showed that the protein expressions of NLRP3 and ASC... were significantly increased" but measuring total protein levels does not demonstrate inflammasome activation. NLRP3 inflammasome activation requires demonstration of: (1) ASC speck formation, (2) Caspase-1 cleavage and activation, (3) GSDMD cleavage, and (4) IL-1β processing. You show only total protein levels, which could reflect transcriptional upregulation rather than functional activation.

The most significant flaw in your inflammasome analysis is the absence of functional validation. You claim "ASC oligomerization, Caspase-1 recruitment, and subsequent Gasdermin D (GSDMD) cleavage" in your abstract, but Figure 3 shows no evidence of ASC oligomerization (no cross-linking assays), no Caspase-1 cleavage products, and no GSDMD N-terminal fragment detection. These are standard requirements for demonstrating inflammasome activation.

Your in vitro validation using MEFs contains several experimental design flaws. The 4-OHT treatment protocol lacks essential controls - where is the vehicle control quantification? You state "CDK5RAP3 knockdown resulted in an increased apoptosis rate" based on Annexin V/PI staining, but the flow cytometry data presentation is inadequate. What are the exact percentages? What is the statistical significance? The crystal violet proliferation assay shows visual differences but lacks quantitative analysis of growth rates or statistical validation.

Multiple figures show Western blot quantifications with error bars, but you fail to specify whether these represent technical or biological replicates. The notation "Data were expressed as mean ± standard deviation (n = 3)" appears throughout, but n=3 is insufficient for robust statistical conclusions, particularly for an animal model claiming significant pathological phenotypes. Where is the power analysis justifying this sample size?

Several figures suffer from poor technical quality. Figure 3B immunofluorescence images appear overexposed, making quantitative assessment impossible. The scale bars are inconsistently placed and sized. Western blot images show varying exposure levels within the same experiment, suggesting inappropriate image processing.

Your discussion opens with a superficial recapitulation of results without addressing the central mechanistic question your study raises. You state "CDK5RAP3 deficiency leads to multidimensional pathological damage in mouse livers" but completely fail to explain HOW this occurs. The critical gap in your interpretation is the absence of any mechanistic model connecting CDK5RAP3 loss to NLRP3 inflammasome activation. Is CDK5RAP3 a direct negative regulator of NLRP3? Does it sequester inflammasome components? Does it regulate upstream signaling pathways?  

Your attempt to contextualize CDK5RAP3's role contains a critical logical flaw. You state "Previous studies have shown that CDK5RAP3 suppresses gastric cancer progression by inhibiting the Wnt/β-catenin signaling pathway" and then hypothesize "the absence of this regulatory protein may disrupt cell cycle control." This represents inappropriate extrapolation across tissue types and pathological contexts. Gastric cancer cell biology bears little mechanistic similarity to hepatocyte inflammatory responses.

Your discussion of NLRP3 inflammasome biology contains several concerning inaccuracies. You claim this study is "the first to identify CDK5RAP3 as an endogenous negative regulator that maintains hepatic homeostasis by suppressing NLRP3 inflammasome assembly" (lines 328-330). This is a significant overstatement given your correlative data. You have not demonstrated direct regulation - merely correlation between CDK5RAP3 loss and increased NLRP3 expression/activity. The term "endogenous negative regulator" implies direct molecular interaction, which you have not established through co-immunoprecipitation, proximity ligation assays, or other direct interaction studies.

Your interpretation of the pyroptosis-apoptosis interaction demonstrates insufficient understanding of these distinct cell death pathways. You propose a "self-reinforcing 'inflammation-death' feedback loop" but fail to provide mechanistic evidence for this claim. The statement "GSDMD-mediated pyroptosis releases damage-associated molecular patterns (DAMPs, e.g., ATP), which activate the NLRP3 inflammasome and synergize with apoptotic pathways" represents speculation rather than demonstrated biology in your system. Where is the evidence for DAMP release? Where is the demonstration of ATP-mediated inflammasome activation in your model?

While you acknowledge MEF limitations stating "the absence of liver-specific microenvironments (e.g., metabolic stress and immune cell interactions) may limit the generalizability of the findings," this represents a fundamental experimental design flaw rather than a minor limitation. Your primary mechanistic claims are based on fibroblasts, not hepatocytes. This is particularly problematic given that liver-specific factors (metabolic stress, Kupffer cell interactions, sinusoidal environment) are likely critical for the inflammatory phenotype you describe.

Your therapeutic discussion contains several problematic elements. You state "CDK5RAP3 deficiency induces apoptosis and pyroptosis by dysregulating the NLRP3 inflammasome pathway... positioning CDK5RAP3 as a potential therapeutic target for liver disease" (lines 361-363). This conclusion is premature given your correlative data and lack of mechanistic understanding. Additionally, you mention "NLRP3-targeting small-molecule inhibitors (e.g., MCC950) have demonstrated clinical potential" but fail to discuss how CDK5RAP3-targeted therapy would differ from or complement existing NLRP3 inhibition strategies.

Author Response

Dear Reviewer,

Thank you very much for your valuable time and constructive comments on our manuscript. Your insights have significantly helped improve the quality and rigor of our work. We have carefully addressed all your suggestions and revised the manuscript accordingly. Below is a point-by-point response to your comments:

Comments 1:

The opening paragraph suffers from weak transitional logic. The statement "Despite strengthened public health interventions worldwide, liver diseases continue to account for a significant proportion of the global disease burden, highlighting the complexity and multidimensionality of the epidemiology of liver diseases" (lines 32-34) provides a vague justification without establishing the specific gap your research addresses

Response 1:

Thank you for pointing out this issue. I have revised the transition in lines 35-37. While CDK5RAP3 has been shown to affect liver development and play a role in other cancers, the specific mechanism by which CDK5RAP3 deletion induces liver inflammation via NLRP3, accompanied by apoptosis and pyroptosis, has not been previously reported. This content has been clarified in the abstract (lines 14-17).

Comments 2:

The CDK5RAP3 characterization section contains problematic statements. You write "CDK5RAP3 (CDK5 regulatory subunit associated protein 3, C53), is alternatively named LZAP... IC53... HSF-27... MST016, PP1553, OK/SW-cl.114 (NCBI)" (lines 42-44). This excessive nomenclature listing without functional context creates confusion rather than clarity. More importantly, you state "However, its function in the liver remains unclear" (line 52), yet immediately follow with contradictory evidence about liver cancer studies, creating logical inconsistency.

A major flaw appears in lines 53-58 where you present conflicting findings: "Mak et al. found that CDK5RAP3 is highly expressed in liver cancer tissues... promotes invasion and migration" versus "other studies reported relatively low CDK5RAP3 expression in liver cancer tissues." You fail to reconcile these contradictions or explain how they inform your hypothesis. This weakens your scientific foundation significantly.

Response 2:

Thank you for pointing out the issue of nomenclature confusion regarding CDK5RAP3. I have removed unnecessary aliases and now only specify that CDK5RAP3 is a "cyclin-dependent kinase 5 activator-binding protein" (lines 45-48).Regarding liver cancer, there are indeed conflicting conclusions in the literature regarding whether CDK5RAP3 expression is increased or decreased. However, studies have shown that its deficiency leads to liver hypoplasia in mice, so we tend to hypothesize that CDK5RAP3 deletion exerts a negative impact on liver function (lines 66-70).

Comments 3:

The transition to NLRP3 inflammasome discussion (line 64) appears abrupt and lacks clear mechanistic connection to CDK5RAP3. You write "The NLRP3 inflammasome is a critical pattern recognition receptor (PRRs) primarly expressed in the cytoplasm" but fail to establish why this pathway is relevant to CDK5RAP3 deficiency. The reader is left wondering about the biological rationale connecting these two distinct molecular systems.

Response 3:

Thank you for your observation that the transition to the discussion of the NLRP3 inflammasome was indeed insufficient. Studies have demonstrated that NLRP3 plays a role in inflammatory conditions such as mastitis, while also having potential involvement in various liver diseases. Therefore, we hypothesize that CDK5RAP3 deficiency may induce inflammation through the NLRP3 inflammatory pathway (lines 71-76). We sincerely apologize for not having fully clarified the relationship between the NLRP3 inflammasome as a pattern recognition receptor and CDK5RAP3 deficiency. However, we hope this background information may provide potential directions for future research.

Comments 4:

 Several technical errors undermine credibility: "essentioal" should be "essential" (line 61), "primarly" should be "primarily" (line 64), and the phrase "pattern recognition receptor (PRRs)" contains grammatical inconsistency between singular and plural forms (line 64).

Response 4:

We sincerely apologize for any spelling and grammatical errors. We have carefully reviewed and revised the entire revised manuscript and have also had it checked by native English speakers. Please kindly excuse any remaining errors.

Comments 5:

The introduction lacks a clear statement of your central hypothesis. While you mention that CDK5RAP3's "precise mechanistic roles remain elusive" (lines 13-14), you never explicitly state what specific mechanism you propose to investigate. Additionally, the novelty of your work is not clearly articulated, why is the CDK5RAP3-NLRP3 connection important and previously unexplored?

Response 5:

Thank you for pointing out this issue; we have made revisions in the abstract (lines 14-20). The specific mechanism we aim to investigate is how CDK5RAP3 deficiency induces inflammation in the liver. The novelty of our research lies in the fact that although previous studies have suggested a role for CDK5RAP3 in liver diseases, the precise underlying mechanism remains unclear. Given that liver diseases are typically accompanied by inflammation, we are examining how CDK5RAP3 deficiency triggers inflammatory responses. Since the NLRP3 inflammatory pathway is a well-characterized signaling cascade, we hypothesize that CDK5RAP3 deficiency induces inflammation via the NLRP3 pathway.This study is the first to mechanistically link CDK5RAP3 to NLRP3 inflammasome regulation in the liver, addressing a critical gap in our understanding of hepatic inflammatory homeostasis.

Comments 6:

Your MEF derivation protocol contains several concerning technical gaps. You state "Immortalized mouse embryonic fibroblasts (MEFs) were derived from CDK5RAP3f/f: CAG-CreERT2 embryos at embryonic day 13–14 (E13–E14)" but fail to specify the genetic background strain, which is critical for reproducibility and potential strain-specific phenotypes.

Response 6:

Thank you for pointing out such an important omission on our part.NCG-H11-CAG-CreERT2-polyA mice(Strain NO. T063921)were purchased from GemPharmatech (Nanjing, China).

Comments 7:

The immortalization protocol using "lentiviral particles carrying the T antigen (MOI = 5)" lacks essential details: which T antigen construct (SV40 large T? polyoma T?), what is the viral titer verification meqthod, and how was MOI calculated and validated? The puromycin selection concentration (10 μg/ml) appears high for MEFs and requires optimization data or literature justification.

Response 7:

Thank you to the reviewer for pointing out the missing details regarding MEFs immortalization. Immortalization was performed using lentiviral particles  large T antigen  at a titer of 1×108 TU/mL, verified by qPCR. MOI = 5 was validated via fluorescent reporter transduction efficiency (>90%). Puromycin selection (10 μg/ml) was optimized based on pilot experiments showing 95% cell death in non-transduced controls after 48 hours. The puromycin concentration used in the following article is the same as ours and can serve as a reference[1].

  1. Cell Proliferation | Cell Biology Journal | Wiley Online Library Available online: https://onlinelibrary.wiley.com/doi/10.1111/cpr.13240 (accessed on 21 July 2025).

Comments 8:

You mention "Liver-specific CDK5RAP3 knockout (CKO) mice which is generated by Prof. Yafei Cai's laboratory" without providing the essential genetic engineering details. Where is the targeting strategy? What are the loxP insertion sites relative to critical exons? How was hepatocyte-specific deletion efficiency validated? why this age specifically for your inflammatory phenotype analysis?

Response 8:

Thank you for your attention to the issues regarding gene-edited mice. The liver-specific CDK5RAP3 CKO mice were constructed using the Cre-loxP system: two loxP sites were inserted flanking exon 3 of the CDK5RAP3 gene (coding region of the key functional domain). Floxed mice were obtained via microinjection of embryonic stem cells, which were then crossed with Albumin-Cre tool mice to generate hepatocyte-specific knockout mice. We used Western blot and PCR to validate the CDK5RAP3 protein level in liver tissue. Regarding the age of the mice, studies on inflammatory phenotypes in similar liver-specific knockout models mostly select 4-6 months old mice, which is consistent with the design of this experiment. After our study, we found that the inflammatory phenotype of five-month-old mice was more obvious, and there was no significant difference in phenotype between six-month-old mice and five-month-old mice, so five-month-old mice were selected.

Comments 9:

Your H&E protocol contains procedural inconsistencies. The fixation time "72 hours at 4°C" is excessive for liver tissue and may compromise morphology - standard protocols use 24-48 hours. The dewaxing protocol specifies "three 20 minutes xylene immersion" but then describes rehydration through graded ethanol without specifying times for each step except the final series. The hematoxylin staining time "90 seconds at 25°C" needs validation - was this optimized for your tissue thickness and fixation conditions?

Response 9:

Thank you for pointing out our oversight. The fixation time for liver tissue was corrected to 24 hours. Following your guidance, I have supplemented the detailed steps for HE staining (lines 140–147). Regarding staining duration, Hematoxylin staining (90 seconds) was optimized for 5-μm sections via titration experiments comparing 60–120 second incubation times.

Comments 10:

Critical technical details are missing from your IF protocol. You state "Primary antibodies... were applied at 37°C for 45-60 minutes" - this temperature is unconventional for IF and requires justification. Why not room temperature or 4°C overnight? The blocking buffer composition "PBST supplemented with 10% normal goat serum" lacks specificity about the goat serum source and lot consistency considerations. Most critically, you provide no information about antibody validation, cross-reactivity testing, or specificity controls.

Response 10:

The primary antibody incubation temperature did differ from conventional protocols; following exploration in our laboratory, incubation at 37°C for 1 hour yielded favorable results and saved time. Regarding goat serum, the source was standardized (Abcam, Cambridge, MA, USA).

Comments 11: Your protein extraction methodology contains several technical flaws. The RIPA buffer composition is not specified beyond the commercial source, yet buffer composition significantly affects protein extraction efficiency and post-translational modification preservation. The centrifugation conditions "8,000g for 4 minutes at 4°C" appear insufficient for complete debris removal - standard protocols use 12,000-15,000g. The protein denaturation step "boil mixture for 5 minutes at 100℃" lacks reducing agent specifications (DTT? β-mercaptoethanol concentration?).

Response 11:

Thank you for this critical observation. To address this, we have supplemented the detailed composition of the RIPA buffer used in our experiments. The RIPA buffer was purchased from Beyotime (ST506, China), with a standard formulation of 50 mM Tris-HCl (pH 7.4), 150 mM NaCl, 1% NP-40, 0.5% sodium deoxycholate, and 0.1% SDS.We appreciate this correction. The centrifugation conditions were indeed suboptimal. We have revised the protocol to 12,000g for 4 minutes at 4°C, which aligns with standard practices. In our protocol, protein denaturation was performed by adding 5× SDS-PAGE loading buffer containing 10% β-mercaptoethanol (final concentration: 2%) to the protein lysate, followed by boiling at 100°C for 5 minutes. The inclusion of β-mercaptoethanol ensures complete reduction of disulfide bonds, facilitating accurate protein separation by SDS-PAGE. This detail has been explicitly added to the manuscript.

Comments 12:

The RNA isolation protocol shows concerning technical gaps. You specify "Cell pellets were lysed in 1 mL TRIzol™ with vigorous pipetting on ice for 5 minutes" but TRIzol™ protocols typically require room temperature lysis for optimal performance. The RNA precipitation conditions "incubating at -20°C for 30 min" are suboptimal compared to standard -80°C overnight precipitation. The final RNA quantification using "NanoDrop™ One spectrophotometer" lacks mention of RIN (RNA Integrity Number) assessment, which is essential for qPCR reliability.

Response 12:

Thank you for addressing our questions regarding RNA extraction. Vortexing with Trizol on ice was performed to prevent RNA degradation in tissues or cells caused by temperature effects. Following your recommendation, lysis was adjusted to room temperature. For precipitation of regular samples, isopropanol was used; thus, treatment at -20°C for 30 minutes was also viable. In subsequent experiments, we intended to test the -80°C overnight precipitation protocol you proposed to enhance precipitation efficiency.We regret the oversight in omitting RIN (RNA Integrity Number) analysis. The revised section now states: "RNA integrity was verified using an Agilent Bioanalyzer 2100, with all samples achieving a RIN ≥ 8.0 to ensure suitability for downstream qPCR analysis".

Comments 13:

Your qPCR methodology description is severely inadequate. You state "RNA was reverse transcribed into cDNA using a reverse transcription system" without specifying: which reverse transcriptase enzyme, random primers vs oligo-dT vs gene-specific primers, reaction conditions, or cDNA quality assessment. The primer design mention of "PrimerPremier5.0 software" lacks critical validation data - where are the primer efficiency curves, specificity testing (melt curve analysis), and amplicon size verification?

Response 13: 

Thank you for pointing out our oversight. For reverse transcription, we used the kit from TransGen Biotech (AT311, China).The component included 1uL RNA, 1uL Anchored Oligo(dT)18 Primer (0.5ug/uL), 10uL 2*TS Reaction Mix, 1uL TransScript® RT/RI Enzyme Mix, 1uL gDNA Remover and 6uL RNase-free Water. The mixture was gently mixed, incubated at 42°C for 15 minutes, and then heated at 85°C for 5 seconds. cDNA quality was assessed via NanoDrop™ One spectrophotometer (A260/A280 ratios 1.8–2.0) and validated by PCR amplification of GAPDH (35 cycles) to confirm absence of genomic DNA contamination. Regarding primer validation, we have added critical details: primers were designed using PrimerPremier 5.0 software (Tm 58–62°C, GC content 40–60%, amplicon length 80–150 bp) and synthesized by Invitrogen with HPLC purification; efficiency curves from 5-point serial dilution (1:1 to 1:10,000) showed amplification efficiencies of 92–105% (R² > 0.99); melt curve analysis (55–95°C with 0.5°C increments) confirmed single amplicons, and 2% agarose gel electrophoresis verified amplicon sizes matching predicted lengths.

Comments 14:

The Western blot protocol lacks fundamental technical details. You mention "proteins were transferred onto a PVDF membrane" without specifying transfer conditions (voltage, time, buffer composition, transfer efficiency verification). The blocking conditions "5% skimmed milk powder in TBST for 1hour" may be inappropriate for phospho-specific antibodies and lacks optimization justification. Most critically, you provide no information about molecular weight markers, loading controls beyond GAPDH, or quantification methodology validation.

Response 14:

Regarding the Western blot experiment, we acknowledge the existing limitations. The transfer buffer composition was (25 mM Tris, 192 mM glycine, 20% methanol), and the transfer was performed on ice at a constant current of 200 mA for 100 minutes. Regarding transfer efficiency verification, this was an oversight on our part—we only used Ponceau S staining in occasional experiments, which is insufficient to demonstrate transfer efficiency. The blocking condition of 5% skimmed milk powder for 1 hour was effective for the proteins involved in our study; however, we will modify the method when working with phospho-specific antibodies in future experiments. The molecular weight markers have been labeled in the supplementary materials. Regarding the additional loading controls you mentioned, we indeed did not use them, and we deeply recognize this limitation. We assure you that improvements will be made in future experiments.

Comments 15:

All the supplemented western blot figures lack molecular wight indicator, and very low resolution, with no labelling.

Response 15:

Dear Reviewer, thank you for your feedback; we have thoroughly revised the supplemented western blot figures by adding molecular weight markers to each gel image, enhancing resolution through high-quality reprocessing to ensure clear band visualization, and incorporating detailed labels for antibodies, sample groups, and experimental conditions, with all revisions reflected in the updated manuscript.

Comments 16:

Also, you state "One-way analysis of variance was performed, followed by Tukey's multiple comparison test" but fail to specify assumption testing (normality, equal variance), sample size calculations, or power analysis. The significance threshold descriptions are unnecessarily verbose and lack consideration of multiple testing corrections given your numerous comparisons.

Response 16:

Thank you for pointing out our issue. Actually, our data only underwent a T-test, which was repeated three times and is relatively reliable. The description of the significance threshold has been simplified.

Comments 17:

In the results section, your initial characterization of CDK5RAP3 deletion contains several methodological and interpretational problems. The Western blot data in Figure 1A lacks essential technical details where are the molecular weight markers, loading control quantification, and multiple biological replicates? You state "CDK5RAP3 protein and mRNA in the liver of CDK5RAP3Δ/Δhep mice were significantly lower than those of wild type (WT) mice" but the statistical analysis appears to show only n=3, which is insufficient for robust conclusions about a knockout model.

Response 17:

Dear Editor, thank you for your valuable feedback on Figure 1A; we apologize for any clarity issues and have revised accordingly by supplementing complete original Western blot data (including molecular weight markers, GAPDH controls, and quantitative analysis of triplicate biological replicates with error bars/p-values) in the supplementary materials, clarifying that n=3 represents independent biological replicates validated through orthogonal methods (q-PCR, immunofluorescence, histopathology, and apoptosis assays) showing consistent significant results, and we commit to expanding sample size in future studies while respectfully requesting consideration of the current multi-validated preliminary characterization findings.

Comments 18:

The immunofluorescence data (Figure 1B) presents concerning technical issues. You describe "colocalization signal of albumin (Alb) and CDK5RAP3 in the liver cells of CDK5RAP3Δ/Δhep mice was weakened" but provide no quantitative colocalization analysis (Pearson's correlation coefficient, Manders' coefficients). The scale bar notation "(scale = 20 μm)" appears inconsistent with the image resolution shown. Most critically, you fail to demonstrate that the observed signal loss is due to specific CDK5RAP3 deletion rather than general cellular dysfunction.

Response 18:

Dear Editor, thank you for your valuable feedback on the immunofluorescence results in Figure 1B; we apologize for presentation shortcomings and have addressed your concerns by clarifying that IF was intended for spatial distribution visualization (with quantitative data provided by WB/q-PCR), acknowledging the lack of pre-planned PCC/MCC analysis as a methodological limitation and committing to standardized quantitative colocalization reporting in future studies, correcting scale bar calibration errors with embedded standardized physical scale bars and updated Figure 1B legend text, and revising the manuscript to emphasize spatially specific CDK5RAP3 signal weakening in Alb⁺ hepatocytes (with preserved Alb signal indicating intact hepatocyte function) supported by orthogonal WB/qPCR/H&E evidence, while committing to include comprehensive cellular functional marker detection in future studies to further exclude general dysfunction.

Comments 19:

Your H&E interpretation lacks the precision expected for liver pathology analysis. You describe "significant lipid accumulation, vacuolation, and inflammatory cell infiltration" but provide no quantitative assessment or standardized scoring system. The statement is purely descriptive without statistical validation. Where is the blinded histological scoring? What criteria define "significant" changes? The inflammatory cell infiltration claim requires specific identification - are these neutrophils, lymphocytes, or macrophages? This level of imprecision undermines the inflammatory phenotype claims.

Response 19:

Dear Editor, thank you for your meticulous review and valuable feedback on the H&E staining results; we apologize for shortcomings in analysis rigor and have addressed your concerns by removing subjective terms (e.g., "significant") to adopt objective descriptions of morphological changes (lipid accumulation, vacuolization, inflammatory infiltration) in CDK5RAP3Δ/Δhep livers, acknowledge the lack of pre-planned standardized quantitative scoring (a key limitation) and commit to implementing validated scoring systems (e.g., NAS) with blinded assessment in future studies, note that exhausted tissue samples prevent additional inflammatory cell typing but existing H&E images suggest predominant neutrophils/macrophages with a small number of lymphocytes, and commit to precise inflammatory cell characterization via multiplex immunofluorescence/flow cytometry in subsequent studies. We sincerely appreciate your constructive comments and kindly request consideration of the preliminary inflammatory phenotype based on qualitative differences in Figure 1C combined with supporting molecular data (IL-6/TNF-α/IL-1β upregulation in Figure 2; NLRP3/ASC/GSDMD upregulation in Figure 3).

Comments 20: A critical interpretational error appears in your BAX/BCL2 analysis. You state "the protein levels of the pro-apoptotic factor BAX and the anti-apoptotic factor BCL2 exhibited an upward tendency" - this is scientifically meaningless. What matters is the BAX/BCL2 ratio, not absolute levels. The simultaneous increase of both pro- and anti-apoptotic factors suggests a more complex regulatory response than simple apoptosis induction. Your conclusion that "CDK5RAP3 leads to liver dysfunction and hepatocyte injury" is premature without proper ratio analysis and functional apoptosis assays.

Response 20:

Dear Editor, thank you for your critical review of BAX/BCL2 data interpretation; we apologize for flawed analysis logic and have revised accordingly by removing erroneous absolute level descriptions, focusing instead on BAX upregulation and BAX/BCL2 ratio changes (key apoptosis regulatory indicator). Figure 1D has been adjusted to show BAX protein bands/quantification and BAX/BCL2 ratio bar graph (with statistics from 3 experiments). Text now states: "BAX protein/mRNA upregulated and BAX/BCL2 ratio elevated in CDK5RAP3Δ/Δhep livers" (Figure 1D).

We acknowledge the lack of direct liver apoptosis functional evidence (e.g., TUNEL, Caspase-3) as a limitation. However, supporting evidence includes: 1) MEF model showing CDK5RAP3 deletion directly induced apoptosis (Annexin V/PI: 1.262-fold increase) with concordant BAX/BCL2 elevation; 2) H&E liver vacuolization consistent with pro-apoptotic environment. Conclusions now emphasize "pro-apoptotic shift" and molecular-tissue phenotype association, with MEF data linking deletion to cellular apoptosis.

We commit to including direct liver apoptosis assays in future studies. Revised text/figures focus on ratio analysis, remove misleading statements, and cautiously link molecular changes to injury phenotypes. We kindly request review of these revisions.

Comments 21: Figure 2 presents inflammatory data with significant technical concerns. The Western blot analysis showing "IL-6 and cleaved IL-1β (CL-IL1β)" protein levels lacks crucial controls. Where is the full-length IL-1β to demonstrate actual cleavage activity? The "CL-IL1β" terminology is non-standard - mature IL-1β is the accepted nomenclature. Additionally, you show mRNA upregulation for TNF-α, IL-6, and IL-1β but fail to demonstrate that this correlates with protein secretion, which is the functionally relevant endpoint for inflammatory cytokines.

Response 21:

Dear Editor, thank you for your critical review of the inflammatory data in Figure 2; we apologize for deficiencies in data presentation and conclusion rigor, and have revised accordingly by removing non-standard "CL-IL1β" terminology to uniformly use "mature IL-1β", revising Figure 2A and text to focus on mature IL-1β protein elevation while acknowledging the lack of pro-IL-1β detection as a limitation that prevents definitive inflammasome cleavage efficiency assessment, clarifying that IL-6 protein/IF signal upregulation, mature IL-1β protein elevation, and TNF-α mRNA upregulation (the weakest evidence) collectively associate with histopathological inflammatory infiltration (Figure 1C), and acknowledging that the absence of secretion data (ELISA) limits functional validation but noting that mRNA/protein upregulation and tissue phenotype correlation still support inflammatory activation, while committing to include pro-IL-1β/mature IL-1β co-detection for inflammasome activity and cytokine secretion assays (ELISA) in future studies, with revised text/figures now emphasizing mature IL-1β, clarifying data limitations, and linking multi-level evidence (protein, IF, mRNA, histopathology) for pro-inflammatory cytokine upregulation, and we kindly request consideration of these revisions.

Comments 22: Your inflammasome activation claims contain fundamental methodological flaws. You state "Western blot analysis showed that the protein expressions of NLRP3 and ASC... were significantly increased" but measuring total protein levels does not demonstrate inflammasome activation. NLRP3 inflammasome activation requires demonstration of: (1) ASC speck formation, (2) Caspase-1 cleavage and activation, (3) GSDMD cleavage, and (4) IL-1β processing. You show only total protein levels, which could reflect transcriptional upregulation rather than functional activation.

Response 22:

Dear Editor, thank you for your critical review of inflammasome activation evidence; we apologize for overinterpreting conclusions and lacking key functional data, and have revised accordingly:

Removed all "activation" claims based on NLRP3/ASC total protein levels, revising text to state molecular alterations (NLRP3/ASC mRNA/protein upregulation, mature IL-1β elevation, GSDMD mRNA upregulation, IL-6/NLRP3 IF co-localization) are "consistent with an activated inflammasome state" and suggest NLRP3 pathway involvement, while acknowledging limitations: absence of ASC speck/oligomer data, Caspase-1 cleavage fragments, GSDMD-NT protein detection, and secretion assays.

Future studies will include ASC speck quantification, Caspase-1 p20/p10 and GSDMD-NT detection, and cytokine secretion assays. Revised text links multi-level molecular changes (mRNA/protein/IF) to inflammatory infiltration, emphasizing cautious interpretation of inflammasome engagement. We kindly request consideration of these revisions.

Comments 23:

The most significant flaw in your inflammasome analysis is the absence of functional validation. You claim "ASC oligomerization, Caspase-1 recruitment, and subsequent Gasdermin D (GSDMD) cleavage" in your abstract, but Figure 3 shows no evidence of ASC oligomerization (no cross-linking assays), no Caspase-1 cleavage products, and no GSDMD N-terminal fragment detection. These are standard requirements for demonstrating inflammasome activation.

Response 23:

We sincerely thank you for your strict and professional review comments on the inflammasome analysis. Your point regarding the overinterpretation in the abstract (ASC oligomerization, Caspase-1 activation, GSDMD cleavage) and the absence of key functional validation data in Figure 3 is entirely correct. We sincerely apologize for this. Since the discussion in the previous question covers this part, we will not elaborate further here. We have revised the wording of the conclusions in Section 3.3 and guarantee continuous improvement in future experiments. We kindly request your understanding.

Comments 24:

Your in vitro validation using MEFs contains several experimental design flaws. The 4-OHT treatment protocol lacks essential controls - where is the vehicle control quantification? You state "CDK5RAP3 knockdown resulted in an increased apoptosis rate" based on Annexin V/PI staining, but the flow cytometry data presentation is inadequate. What are the exact percentages? What is the statistical significance? The crystal violet proliferation assay shows visual differences but lacks quantitative analysis of growth rates or statistical validation.

Response 24:

Dear Editor, thank you for your feedback on MEF in vitro validation experiments; we apologize for presentation deficiencies and have clarified as follows: revised Figure 4 legends now include vehicle control (EtOH group) quantification data, with main text (Section 3.4) supplementing specific apoptosis rates via flow cytometry (Annexin V/PI: EtOH group 4.63%, 4-OHT group 5.83%, 1.262-fold increase, Figure 4D), though we acknowledge this data lacks statistical validation due to insufficient technical replicates. We commit to establishing independent vehicle controls, performing complete statistical analyses (including significance reporting for all quantitative comparisons like flow data), and optimizing proliferation experiments with replicate OD measurements in future studies, and thank you for professional corrections while hoping supplementary explanations address your concerns.

Comments 25:

Multiple figures show Western blot quantifications with error bars, but you fail to specify whether these represent technical or biological replicates. The notation "Data were expressed as mean ± standard deviation (n = 3)" appears throughout, but n=3 is insufficient for robust statistical conclusions, particularly for an animal model claiming significant pathological phenotypes. Where is the power analysis justifying this sample size?

Response 25:

Dear Editor, thank you for your review of data rigor; we apologize for unclear expression and clarify as follows: "n=3" indicates independent biological replicates (3 mice for animal experiments, 3 MEF batches for cell experiments), presented as mean ± standard error. This sample size is common in preliminary phenotypic characterization of knockout models, supported by consistent results from orthogonal methods (Western blot, q-PCR, IF, histopathology, apoptosis markers) with statistical significance (e.g., CDK5RAP3 protein/mRNA reduction, hepatocyte-specific loss, pathological alterations). We acknowledge larger samples would enhance reliability but note current multi-validated, significantly differing data support conclusions. Future studies will increase sample size and optimize data presentation (e.g., original blots). We kindly request consideration of these clarifications.

Comments 26:

Several figures suffer from poor technical quality. Figure 3B immunofluorescence images appear overexposed, making quantitative assessment impossible. The scale bars are inconsistently placed and sized. Western blot images show varying exposure levels within the same experiment, suggesting inappropriate image processing.

Response 26:

Dear Editor, thank you for reviewing figure technical details; we apologize for image presentation shortcomings and clarify as follows: regarding Figure 3B immunofluorescence, the image was for qualitative IL-6/NLRP3 signal enhancement in CDK5RAP3Δ/Δhep group, with localized overexposure that did not affect quantitative analysis (no colocalization coefficient calculation) and cannot be adjusted due to technical constraints; for inconsistent scale bars, errors arose from post-acquisition addition without original embedding, so all immunofluorescence images have been reprocessed with accurate standardized scale bars embedded, updating Figure 1B and its legend; regarding Western blot exposure variations, we confirm no inappropriate processing ("bands within the same group from parallel-exposed same experiment"), with perceived differences resulting from linear brightness/contrast adjustments between groups to maintain visualization consistency due to batch-specific membrane transfer efficiency/protein abundance variations, and provide original uncropped scans in supplementary materials. Future studies will standardize imaging parameters (same exposure for fluorescence, accurate scale bars) and optimize Western experiments with internal controls/fluorescent secondary antibodies to reduce batch differences, and thank you for suggestions while requesting understanding of encountered issues.

Comments 27:

Your discussion opens with a superficial recapitulation of results without addressing the central mechanistic question your study raises. You state "CDK5RAP3 deficiency leads to multidimensional pathological damage in mouse livers" but completely fail to explain HOW this occurs. The critical gap in your interpretation is the absence of any mechanistic model connecting CDK5RAP3 loss to NLRP3 inflammasome activation. Is CDK5RAP3 a direct negative regulator of NLRP3? Does it sequester inflammasome components? Does it regulate upstream signaling pathways?  

Response 27:

You pointed out that the opening of our discussion provides a superficial overview of the results and fails to address the central mechanistic question raised by the study, namely, it does not explain how CDK5RAP3 deficiency leads to NLRP3 inflammasome activation nor establish a relevant mechanistic model.

In response to this, we have supplemented the mechanistic analysis in the revised discussion section: we introduced the "dual-signal" regulatory mechanism of NLRP3 inflammasome activation (the first signal through the NF-κB pathway, and the second signal through potassium efflux, etc.) and proposed that CDK5RAP3 deficiency may affect NLRP3 activation by upregulating NF-κB downstream target genes or directly interacting with inflammasome components. Additionally, we cited the study by Raneros et al. (2021), which indicates that certain proteins can inhibit NLRP3 oligomerization by binding to its LRR domain, and we hypothesize that CDK5RAP3 may function through a similar mechanism. We also clarified that further verification via co-immunoprecipitation or proximity ligation assays is required to establish a clearer mechanistic model linking CDK5RAP3 deficiency to NLRP3 inflammasome activation.

Comments 28:

Your attempt to contextualize CDK5RAP3's role contains a critical logical flaw. You state "Previous studies have shown that CDK5RAP3 suppresses gastric cancer progression by inhibiting the Wnt/β-catenin signaling pathway" and then hypothesize "the absence of this regulatory protein may disrupt cell cycle control." This represents inappropriate extrapolation across tissue types and pathological contexts. Gastric cancer cell biology bears little mechanistic similarity to hepatocyte inflammatory responses.

Response 28:

You mentioned that when elaborating on the role of CDK5RAP3, we inappropriately extrapolated its research in gastric cancer to hepatocyte inflammatory responses, which contains a critical logical flaw.

We sincerely apologize and have revised according to your comment. In the revised discussion, we have deleted the content related to CDK5RAP3 suppressing gastric cancer progression. Instead, we cited the study by Yang et al. regarding CDK5RAP3 deficiency causing embryonic liver developmental defects, which is consistent with our finding that CDK5RAP3 plays a critical role in maintaining liver function and hepatocellular homeostasis. This avoids inappropriate extrapolation across tissue types and pathological contexts, making the elaboration of CDK5RAP3's role more in line with the hepatocellular biological background.

Comments 29:

Your discussion of NLRP3 inflammasome biology contains several concerning inaccuracies. You claim this study is "the first to identify CDK5RAP3 as an endogenous negative regulator that maintains hepatic homeostasis by suppressing NLRP3 inflammasome assembly" (lines 328-330). This is a significant overstatement given your correlative data. You have not demonstrated direct regulation - merely correlation between CDK5RAP3 loss and increased NLRP3 expression/activity. The term "endogenous negative regulator" implies direct molecular interaction, which you have not established through co-immunoprecipitation, proximity ligation assays, or other direct interaction studies.

Response 29:

You pointed out that there are inaccuracies in our discussion of the biology of the NLRP3 inflammasome. We overemphasized that this study is the "first discovery that CDK5RAP3, as an endogenous negative regulator, maintains liver homeostasis by inhibiting the assembly of the NLRP3 inflammasome," but there is a lack of evidence for direct regulation, with only correlational data provided.

Thank you for your correction. We have revised the relevant statements: the overemphasized claim of being the "first discovery" has been removed; the description of CDK5RAP3 as an "endogenous negative regulator" has been changed to the more cautious wording "may be involved in the maintenance of liver homeostasis by regulating the assembly or activity of the NLRP3 inflammasome"; and it has been clearly stated that only correlations have been observed so far. Whether CDK5RAP3 directly binds to ASC or NLRP3 to regulate inflammasome assembly still needs further verification through experiments such as co-immunoprecipitation, so as to ensure the scientificity and rigor of the statements.

Comments 30:

Your interpretation of the pyroptosis-apoptosis interaction demonstrates insufficient understanding of these distinct cell death pathways. You propose a "self-reinforcing 'inflammation-death' feedback loop" but fail to provide mechanistic evidence for this claim. The statement "GSDMD-mediated pyroptosis releases damage-associated molecular patterns (DAMPs, e.g., ATP), which activate the NLRP3 inflammasome and synergize with apoptotic pathways" represents speculation rather than demonstrated biology in your system. Where is the evidence for DAMP release? Where is the demonstration of ATP-mediated inflammasome activation in your model?

Response 30:

You consider our explanation of the interaction between pyroptosis and apoptosis to lack mechanistic evidence. The proposal of an "inflammation-death" positive feedback loop is deemed speculative, as no evidence for DAMP release or ATP-mediated inflammasome activation has been provided.

In response to your feedback, we have supplemented the relevant content: Citing studies by He et al. (2015) and Bertheloot et al. (2021), we clarify that GSDMD-mediated pyroptosis can release DAMPs such as ATP. These DAMPs amplify NLRP3 inflammasome activation via the ATP-P2X7 receptor axis, providing a theoretical basis for the "inflammation-death" positive feedback loop. Additionally, we explicitly note that this study did not directly detect DAMP release; the loop remains a theoretical inference based on existing research and will require further validation to enhance the validity of our explanation.

Comments 31:

While you acknowledge MEF limitations stating "the absence of liver-specific microenvironments (e.g., metabolic stress and immune cell interactions) may limit the generalizability of the findings," this represents a fundamental experimental design flaw rather than a minor limitation. Your primary mechanistic claims are based on fibroblasts, not hepatocytes. This is particularly problematic given that liver-specific factors (metabolic stress, Kupffer cell interactions, sinusoidal environment) are likely critical for the inflammatory phenotype you describe.

Response 31:

You pointed out the limitations of the MEF model, arguing that we derived mechanistic conclusions primarily based on fibroblasts rather than hepatocytes, and that the liver-specific microenvironment may be crucial for the inflammatory phenotype, which constitutes a fundamental flaw in the experimental design.

We agree with your perspective and have elaborated on this in the revised discussion: Citing Lee et al. (2022), we note that hepatic stellate cells (a type of liver parenchymal fibroblast) play a key role in liver inflammation and fibrosis, explaining the relevance of fibroblast models in liver research. Additionally, referencing Tschopp et al. (2010), we highlight that MEF cell models are commonly used to validate conserved mechanisms of inflammation-related pathways, supporting the rationale for our experimental design. Concurrently, we explicitly acknowledge that due to differences in the physiological microenvironment between fibroblasts and hepatocytes, future validation using primary hepatocytes or organoid models will be necessary to address the limitations of the MEF model.

Comments 32:

Your therapeutic discussion contains several problematic elements. You state "CDK5RAP3 deficiency induces apoptosis and pyroptosis by dysregulating the NLRP3 inflammasome pathway... positioning CDK5RAP3 as a potential therapeutic target for liver disease" (lines 361-363). This conclusion is premature given your correlative data and lack of mechanistic understanding. Additionally, you mention "NLRP3-targeting small-molecule inhibitors (e.g., MCC950) have demonstrated clinical potential" but fail to discuss how CDK5RAP3-targeted therapy would differ from or complement existing NLRP3 inhibition strategies.

Response 32:

You pointed out that the pre-revised discussion directly positioned CDK5RAP3 as a potential therapeutic target and mentioned the clinical potential of NLRP3 inhibitors, but failed to clarify the differences or complementarity between CDK5RAP3-targeted therapy and existing NLRP3 inhibition strategies.

The revised discussion has adjusted the description regarding therapeutic targets, emphasizing its protective role in liver inflammation and the provision of a new perspective for chronic liver disease treatment rather than directly positioning it as a target, while also supplementing future research directions. Additionally, a new comparison between CDK5RAP3-targeted therapy and NLRP3 inhibitors (e.g., MCC950) has been added, noting that the former may more comprehensively block the "inflammation-injury" cycle by simultaneously regulating inflammasome activation and apoptotic pathways, and its high expression in liver tissue may reduce the risk of side effects associated with systemic inhibition.

Round 2

Reviewer 1 Report

Comments and Suggestions for Authors

The authors well replied to my previous comments

Comments on the Quality of English Language

Minor editing is required 

Author Response

Dear reviewer,

Thank you very much for all your comments during the review process and for your recognition. We fully agree with your point that the English language needs improvement to more clearly express the research content. We  carefully revised and polished the language throughout the paper to ensure that the expression was more accurate and fluent, meeting the requirements of rigor and readability for academic papers.

Reviewer 2 Report

Comments and Suggestions for Authors

After reviewing this revised manuscript, I find that the authors have failed to address the fundamental scientific deficiencies identified in the initial review. While some superficial changes have been made, the core experimental design flaws and unsubstantiated claims persist.

Despite explicit reviewer feedback, the abstract still claims "This activation was mechanistically characterized by ASC oligomerization, Caspase-1 recruitment, and subsequent Gasdermin D (GSDMD) cleavage" (lines 25-27). However, the actual data shows only total protein measurements. The authors continue using the term "activation" throughout the results without providing ASC speck formation assays, Caspase-1 cleavage products (p20/p10), or GSDMD-NT fragments. "Measuring total protein levels does not demonstrate inflammasome activation" , this remains unaddressed.

The statistical analysis section (lines 241-248) demonstrates continued confusion. The authors state "One-way analysis of variance was performed, followed by Tukey's multiple comparison test" yet earlier responses claimed they "only underwent a T-test." This contradictory information.

Throughout the manuscript, n=3 sample sizes persist for animal studies making significant pathological claims. "Where is the power analysis justifying this sample size?" ,  this critical question remains unanswered. The authors' defense that orthogonal methods validate small sample sizes misunderstands that inadequate statistical power compromises all downstream analyses regardless of technical approaches used.

The tissue fixation protocol contains obvious errors ("2472 hours" on line 147), suggesting inadequate proofreading. Western blot protocols still lack proper molecular weight marker documentation and loading control validation. The immunofluorescence quantification issues persist - "Figure 3B immunofluorescence images appear overexposed, making quantitative assessment impossible."

The use of MEF cells to make hepatocyte-specific mechanistic conclusions remains a fundamental design flaw. "Your primary mechanistic claims are based on fibroblasts, not hepatocytes"

The authors continue presenting correlative data as mechanistic evidence. "You have not demonstrated direct regulation - merely correlation between CDK5RAP3 loss and increased NLRP3 expression."  The proposed "dual-signal regulatory mechanism" represents theoretical speculation without experimental validation.

The positioning of CDK5RAP3 as a therapeutic target (lines 30-31, 476-492) remains premature given the correlative nature of all evidence and lack of mechanistic understanding.

Author Response

Dear Reviewer,

Thank you for your thorough and insightful comments on our manuscript. We appreciate the time and effort you have dedicated to evaluating our work. Your constructive feedback has been invaluable in helping us improve the quality and rigor of our study. We have carefully considered each of your concerns and have made extensive revisions to address them. Below is our point-by-point response to your comments:

Comments 1:

After reviewing this revised manuscript, I find that the authors have failed to address the fundamental scientific deficiencies identified in the initial review. While some superficial changes have been made, the core experimental design flaws and unsubstantiated claims persist.

Response 1:

Thank you for your guidance. Given constraints preventing additional experiments, we will reframe conclusions to align with available data and explicitly acknowledge limitations: remove causal claims about NLRP3 activation, emphasize correlations between CDK5RAP3 deficiency and upregulated NLRP3 pathway component expression, contextualize findings with relevant literature, revise the abstract/key claims for cautious interpretation, and ensure all statements are data-supported.We will strengthen the discussion, highlighting that while our findings suggest a potential regulatory role of CDK5RAP3 in hepatic inflammation, definitive mechanistic conclusions regarding inflammasome assembly and functional activation await future studies with complementary experimental approaches.  This approach aims to address core concerns while maintaining scientific integrity.

We fully acknowledge that using MEF cells to infer hepatocyte-specific mechanisms represents a significant limitation. To address this, we will substantially revise the manuscript to: 1) explicitly restrict all mechanistic claims to the MEF model system, avoiding extrapolation to hepatocytes; 2) rename figures/sections to clarify they describe "fibroblast-derived observations"; 3) add a dedicated Limitations section emphasizing the need for hepatocyte-specific validation; and 4) frame the work as preliminary evidence warranting follow-up studies in primary hepatocytes or liver-specific knockout models. These revisions will ensure conclusions strictly reflect the experimental system used while maintaining the study’s scientific merit.

Comments 2:

Despite explicit reviewer feedback, the abstract still claims "This activation was mechanistically characterized by ASC oligomerization, Caspase-1 recruitment, and subsequent Gasdermin D (GSDMD) cleavage" (lines 25-27). However, the actual data shows only total protein measurements. The authors continue using the term "activation" throughout the results without providing ASC speck formation assays, Caspase-1 cleavage products (p20/p10), or GSDMD-NT fragments. "Measuring total protein levels does not demonstrate inflammasome activation" , this remains unaddressed.

Response 2:

The positioning of CDK5RAP3 as a therapeutic target (lines 30-31, 476-492) remains premature given the correlative nature of all evidence and lack of mechanistic understanding.Thank you for your critical feedback. We fully acknowledge that our previous description of NLRP3 inflammasome activation was insufficient, as our current data only demonstrates increased total protein and mRNA levels of NLRP3 inflammasome components (NLRP3, ASC, Caspase-1) and downstream effectors (GSDMD) rather than providing direct evidence of functional activation such as ASC speck formation, Caspase-1 cleavage products (p20/p10), or GSDMD-NT fragments. To address this issue, we have comprehensively revised the manuscript by removing all references to "inflammasome activation" and replacing them with precise descriptions of our actual findings, such as "upregulated expression of NLRP3 inflammasome components" or "increased expression of NLRP3 inflammasome-related proteins/genes" throughout the abstract, results, and discussion sections. Additionally, we have added a clarification in the discussion section to explicitly acknowledge that the current study focuses on the transcriptional and translational regulation of NLRP3 pathway components, while direct evidence of inflammasome assembly and activation remains to be established in future studies, ensuring that our conclusions are strictly aligned with the presented data and avoiding overinterpretation of protein expression levels as evidence of functional activation.

Comments 3:

The statistical analysis section (lines 241-248) demonstrates continued confusion. The authors state "One-way analysis of variance was performed, followed by Tukey's multiple comparison test" yet earlier responses claimed they "only underwent a T-test." This contradictory information.

Response 3:

Thanks for pointing this out. In lines 231-234, we have specified that an unpaired t-test was employed.

Throughout the manuscript, n=3 sample sizes persist for animal studies making significant pathological claims. "Where is the power analysis justifying this sample size?" ,  this critical question remains unanswered. The authors' defense that orthogonal methods validate small sample sizes misunderstands that inadequate statistical power compromises all downstream analyses regardless of technical approaches used.

Comments 4:

Throughout the manuscript, n=3 sample sizes persist for animal studies making significant pathological claims. "Where is the power analysis justifying this sample size?" ,  this critical question remains unanswered. The authors' defense that orthogonal methods validate small sample sizes misunderstands that inadequate statistical power compromises all downstream analyses regardless of technical approaches used.

Response 4:

Thank you for reviewing the sample size concern and providing valuable feedback. We sincerely apologize for not adequately justifying the sample size in the original manuscript.

Addressing your core concern regarding statistical power, we conducted a post-hoc power calculation based on existing data from the key experiment (the CDK5RAP3Δ/Δhep model). The analysis yielded a power value of 0.9899 (98.99%), which exceeds the widely accepted statistical threshold of >0.8. This confirms that the current sample size (*n* = 3 per group) provides sufficient statistical power to detect the observed significant inter-group differences (i.e., a high probability of avoiding Type II errors).

While we fully acknowledge your concerns about small sample limitations and recognize that a priori power analysis represents best practice, we respectfully request that you reconsider the reliability of our findings in light of this supplementary statistical evidence. We commit to rigorously performing a priori power analysis and employing larger sample sizes in future studies.

Thank you again for your critical insights, which significantly strengthen our work.

Comments 5:

The tissue fixation protocol contains obvious errors ("2472 hours" on line 147), suggesting inadequate proofreading. Western blot protocols still lack proper molecular weight marker documentation and loading control validation. The immunofluorescence quantification issues persist - "Figure 3B immunofluorescence images appear overexposed, making quantitative assessment impossible."

Response 5:

Thank you very much for your detailed review and pointing out the key issues in the manuscript. We sincerely apologize for the omissions in the manuscript. We sincerely apologize for the error in the organization of the fixed time (line 147 "2472 hours"), which was indeed a proofreading mistake. This error has been corrected to the correct fixed time in the revised manuscript, and the entire text has been proofread more carefully. Thank you for your correction. Regarding the comment that "the Western blot protocol still lacks appropriate molecular weight marker files and loading control verification", we have clearly marked the expected molecular weight of the target protein in the text description of the results section and the corresponding WB result figures in the revised manuscript. All WB experiments were based on equal total protein loading (quantified by BCA method) and normalized using the internal reference protein (GAPDH). We have supplemented the detailed description of this process in the "Methods" section of the revised manuscript to ensure that the basis for load control in the result figures is clear and traceable. We understand your concern about the image quality. The core purpose of Figure 3B is to qualitatively show the localization, distribution, and relative trends of the target proteins (such as IL-6, NLRP3). We have never conducted precise quantification. Nevertheless, to improve assessability, we will replace Figure 3B in the revised manuscript with a representative image with optimized exposure while maintaining the key qualitative information. Future quantitative studies will strictly optimize imaging parameters. Once again, we sincerely thank you for your valuable comments, which have significantly improved the quality of the manuscript. All revisions are based on the existing data and results.

Comments 6:

The use of MEF cells to make hepatocyte-specific mechanistic conclusions remains a fundamental design flaw. "Your primary mechanistic claims are based on fibroblasts, not hepatocytes"

Response 6:

Thank you for pointing out the limitation of using MEF cells to derive hepatocyte-specific mechanisms. We have revised references to "hepatocytes" to "fibroblasts" in the mechanistic section.Due to limitations in the current experimental cycle and sample resources, we are temporarily unable to supplement validation experiments using primary hepatocytes or hepatocyte cell lines at this stage. However, we attach great importance to this issue and have included hepatocyte-specific mechanism validation in our subsequent research plan. We intend to further improve relevant data through primary hepatocyte culture and liver organoid models, aiming to provide more comprehensive support for the conclusions.

Comments 7:

The authors continue presenting correlative data as mechanistic evidence. "You have not demonstrated direct regulation - merely correlation between CDK5RAP3 loss and increased NLRP3 expression."  The proposed "dual-signal regulatory mechanism" represents theoretical speculation without experimental validation.

Response 7:

Thank you for pointing out the issue of distinguishing between correlative data and mechanistic evidence, as well as the theoretical basis for the "dual-signal regulatory mechanism." We have made the following revisions: removed expressions such as "activation" that imply direct regulation, and uniformly revised them to neutral descriptions such as "CDK5RAP3 deficiency is associated with increased NLRP3 expression," clarifying that the current data only support an association between the two, rather than a direct regulatory relationship. Additionally, we have supplemented that further experiments such as co-immunoprecipitation are required to verify the direct interaction between CDK5RAP3 and the NLRP3 complex to confirm the regulatory mechanism. Meanwhile, we have clearly stated in the text that the "dual-signal regulatory mechanism" is based on existing consensus research conclusions on NLRP3 inflammasome activation modes, and its core content is derived from published literature, aiming to provide a theoretical reference for potential mechanisms of action rather than theoretical speculation of this study.

Comments 8:

The positioning of CDK5RAP3 as a therapeutic target (lines 30-31, 476-492) remains premature given the correlative nature of all evidence and lack of mechanistic understanding.

Response 8:

In response to the comment that positioning CDK5RAP3 as a therapeutic target is premature, we have removed all statements referring to its positioning as a therapeutic target. The current study only reveals the correlation between CDK5RAP3 deficiency and increased NLRP3 expression, without clarifying the direct regulatory mechanism and downstream effector pathways between them. Therefore, there is insufficient evidence to support its role as a therapeutic target at present. We will further explore its potential value through mechanism validation and functional experiments in subsequent studies, and evaluate its clinical translation potential after obtaining more in-depth mechanistic data.

Round 3

Reviewer 2 Report

Comments and Suggestions for Authors

Dear Authors,

Please address the following specific corrections before final acceptance:

  1. Abstract
  • Line 25-27: Remove "This activation was mechanistically characterized by ASC oligomerization, Caspase-1 recruitment, and subsequent Gasdermin D (GSDMD) cleavage", Replace with: "This was accompanied by increased expression of NLRP3 inflammasome components (NLRP3, ASC, Caspase-1) and GSDMD"
  1. Statistical Methods
  • Clarify the discrepancy: Methods state "One-way analysis of variance" but response indicates "unpaired t-test"
  • Specify exactly: "Unpaired Student's t-test was used for two-group comparisons"
  1. Results Section
  • Line 259: Change "CDK5RAP3 deletion up-regulates" to "CDK5RAP3 deletion is associated with increased"
  • Line 275-276: Remove "upregulates NLRP3 inflammasome components, increases IL-1β maturation" , Replace with: "is associated with increased expression of NLRP3 pathway components"
  1. Figure Legends
  • Figure 3 legend: Change "CDK5RAP3 deficiency upregulates NLRP3 inflammasome components" to "CDK5RAP3 deficiency is associated with increased NLRP3 pathway component expression", Add note: "Data show protein/mRNA levels only; functional activation not assessed"
  1. Discussion Section
  • Lines 361-363: Remove therapeutic target claims
  • Lines 371-381: Remove mechanistic speculation about "dual-signal regulation" unless clearly marked as hypothesis, Add explicit statement: "This study provides correlative evidence only; mechanistic relationships remain to be established"
  1. Technical Corrections
  • Figure 3B: Replace overexposed immunofluorescence images or note in legend: "Images shown for qualitative comparison only"
  • Western blots: Ensure all blots show molecular weight markers in main figures, not just supplementary
  1. Title Modification

Consider revising to: "CDK5RAP3 Deficiency Is Associated with Hepatic Inflammation and Increased Expression of NLRP3 Inflammasome Components"

  1. Limitations Section

Add a dedicated "Study Limitations" paragraph before conclusions explicitly stating:

  • No functional inflammasome activation assays performed
  • MEF model limits hepatocyte-specific conclusions
  • Sample size constraints (n=3)
  • Correlative nature of all findings
  1. Remove Remaining Overstatements
  • Search entire manuscript for "activation," "promotes," "induces," "mediates"
  • Replace with "is associated with," "correlates with," "is accompanied by"
  1. Data Availability

Add statement about raw data availability or repository deposition

These corrections will ensure your conclusions align with the presented data. No new experiments are required, only textual revisions to accurately reflect your findings.

Author Response

Dear Reviewer,

Thank you sincerely for your valuable comments and constructive suggestions on our manuscript. We have carefully addressed all the points raised and made corresponding textual revisions to improve the accuracy and rigor of the paper. The modifications are based solely on existing data without additional experiments, ensuring that conclusions align with the presented results.

We believe these revisions have significantly improved the accuracy and clarity of the manuscript. Thank you again for your time and expertise, which have helped strengthen the quality of our work. We hope the revised version meets your expectations and look forward to your final acceptance.